# Lever LM: Configuring In-Context Sequence to Lever Large Vision Language Models

**Xu Yang**[1,2]*, **Yingzhe Peng**[1,2], **Haoxuan Ma**[1,2], **Shuo Xu**[1,2],
**Chi Zhang**[3], **Yucheng Han**[4], **Hanwang Zhang**[4]

[1] Southeast University
[2] Key Laboratory of New Generation Artificial Intelligence Technology &
Its Interdisciplinary Applications, (Southeast University),Ministry of Education
[3] Westlake University
[4] Nanyang Technological University
{xuyang_palm, yingzhe.peng, haoxuan-ma, xushuo}@seu.edu.cn
chizhang@westlake.edu.cn, yucheng002@e.ntu.edu.sg, hanwangzhang@ntu.edu.sg

## Abstract

As Archimedes famously said, "Give me a lever long enough and a fulcrum on which to place it, and I shall move the world", in this study, we propose to use a tiny Language Model (LM), *e.g.*, a Transformer with 67M parameters, to lever much larger Vision-Language Models (LVLMs) with 9B parameters. Specifically, we use this tiny **Lever-LM** to configure effective in-context demonstration (ICD) sequences to improve the In-Context Learinng (ICL) performance of LVLMs. Previous studies show that diverse ICD configurations like the selection and ordering of the demonstrations heavily affect the ICL performance, highlighting the significance of configuring effective ICD sequences. Motivated by this and by re-considering the the process of configuring ICD sequence, we find this is a mirror process of human sentence composition and further assume that effective ICD configurations may contain internal statistical patterns that can be captured by Lever-LM. Then a dataset with effective ICD sequences is constructed to train Lever-LM. After training, given novel queries, new ICD sequences are configured by the trained Lever-LM to solve vision-language tasks through ICL. Experiments show that these ICD sequences can improve the ICL performance of two LVLMs compared with some strong baselines in Visual Question Answering and Image Captioning, validating that Lever-LM can really capture the statistical patterns for levering LVLMs. The code is available at https://github.com/ForJadeForest/Lever-LM.

## 1 Introduction

With the escalation in model size and training data [1–6], Large Language Models (LLMs) emerge the ability of In-Context Learning (ICL) [7–9]. ICL, akin to few-shot learning [10–12], utilizes a few exemplary In-Context Demonstrations (ICDs) to adapt LLMs to new tasks without gradient updates. This achievement in NLP has inspired researchers to similarly enhance Large Vision-Language Models (LVLMs) with ICL capabilities [13, 14]. However, just as in NLP, the effectiveness of ICL in LVLMs is significantly influenced by the configurations of ICDs, such as their selection and ordering [15–23]. Recent studies [24–26] have shown that this sensitivity in LVLMs is further exacerbated by the multimodal combinatorial complexity of vision and language data.

In NLP, researchers employ various strategies to optimize in-context sequences to improve ICL performance, including retrieving representative examples as the ICDs [15, 27, 28] and re-ordering these ICDs based on specific principles [29, 21]. While these methods have shown improvements,

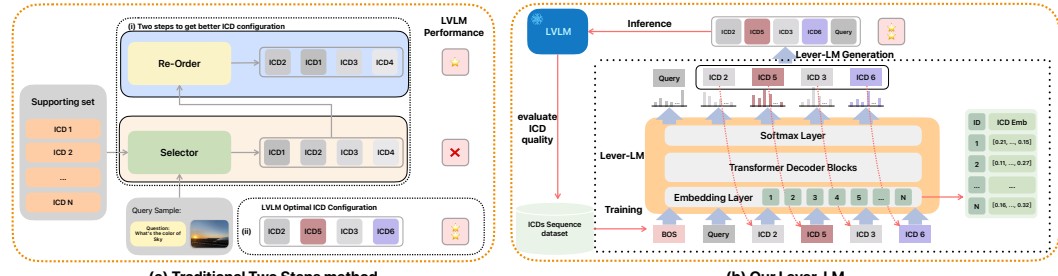

Figure 1: (a) The traditional ICD configuration methods separately select and order the ICDs, leading to sub-optimal ICL performance. (b) Our Lever-LM enables the step-by-step generation of ICD configurations and simultaneously considers the selection of ICDs and the ordering of ICD sequences.

their application remains largely confined to NLP and is less explored in the vision-language domain. Moreover, as shown in Fig. 1(a), the independent operations of retrieval and reordering often result in sub-optimal outcomes. A critical reconsideration of the ICD sequence generation reveals that configuring an optimal ICD sequence should be a coherent process. Instead of independently selecting and re-ordering, each ICD should be chosen conditionally based on the previous ICDs. This mirrors the sequential nature of human sentence composition, where each word is sequentially selected to ensure overall fluency. Such fluency can be characterized by temporal statistical patterns, which allows the design of statistical learning methods to model and learn from data, where Language Model is one typical technique that demonstrates the effectiveness. This analogy supports the hypothesis that optimal ICD sequences may contain inherent temporal statistical patterns.

Although not explicitly stated, some previous studies in NLP work toward this direction by calculating statistical metrics like the perplexity [30] or the entropy [31, 32] to discover what statistical characteristics a good prompt should have. However, the requirement to know the probability of each token when calculating these metrics limits their applicability in VL. This is because recently proposed LVLMs [13, 14, 33] use continuous image patches rather than the tokenized discrete elements as the vision input, rendering these techniques used in NLP inapplicable in VL and the following two questions remain unaddressed: (1) whether effective VL ICDs exhibit certain statistical patterns and (2) whether such patterns can be leveraged to compose new ICD sequences for a given query. This study aims to address these questions. Specifically, we employ a tiny Language Model, *e.g.*, a Transformer, to capture the inherent statistic patterns. This tiny LM is named as "**Lever-LM**" since it can lever/control a much larger VLM by composing suitable ICD sequence. Compared with the classic LM, the only difference is that the vocabulary of Lever-LM consists not of standard words, but of examples from the supporting set that will be used as ICDs.

Fig. 1 (b) shows the training pipeline for Lever-LM. Initially, a "ground-truth" dataset is constructed to indicate which examples or their orders can form good ICD sequences. Specifically, we employ a frozen LVLM to evaluate if an ICD sequence facilitates accurate predictions for a given query, *e.g.*, answering questions correctly or generating appropriate captions. [1] Lever-LM is then trained to concurrently learn the selection and ordering of ICDs, streamlining the process by eliminating the need for two separate stages typical of previous methods. Our experiments, conducted with two LVLMs—Open-Flamingo [34] and IDEFICS [14]—on classic VL tasks—Image Captioning and Visual Question Answering— demonstrate that Lever-LM surpasses several strong baselines, including those that retrieve ICDs based on image similarity. These results confirm that effective ICD sequences contain inherent temporal statistical patterns and such patterns can be learned for composing new ICD sequences for test queries.

Besides the above-mentioned advantages, Lever LM emerges two interesting abilities. First, it has strong length extrapolation ability, *e.g.*, when trained on a dataset with only 2-shot ICDs, Lever LM can generate 4 or more-shot ICDs that outperform several strong baselines. Second, Lever LM can construct a "golden" ICD sequence of 8 predetermined ICDs in a fixed order. This sequence can be uniformly applied across different test queries to assist LVLM in label generation, thereby reducing the computational overhead for configuring new ICD sequences for each query. Experiments in IC/VQA tasks show that "golden" ICD sequence achieves 6.91/1.24 improvements compared to

---

[1]Note: This method requires ground-truth labels and thus can not be used at the test stage.

a strong baseline. In addition, we use exhaustive ablations, including applying different ways to construct training set and changing the architecture of Lever-LM, to discover which factors and analyze why they will affect the ICL performance.

## 2 Related Work

**Models with In-Context Learning Ability.** Prompt engineering enables Large Language Models (LLMs) to address downstream tasks without the need for fine-tuning [35, 1, 36]. A variant, ICL, enhances this ability by constructing prompts with a few examples. This has been demonstrated in LLMs such as GPT-3 [1], LLaMA [5], and MPT [37]. Recently, witnessing such success in NLP, the VL domain has also developed numerous LVLMs with prompt engineering abilities [38, 33, 39, 13, 40–44] and ICL ability like [40], Flamingo [13], and IDEFICS [14] Among them, we use Flamingo and IDEFICS as the LVLMs to explore the effectiveness of Lever-LM since they have stronger and more robust ICL ability by using better language encoders and more training data[2].

**Configuring In-Context Demonstrations.** Although ICL assists LLMs in better adapting to downstream tasks, its performance is highly sensitive to the selection [15, 19, 20] and ordering [21–23] of ICDs. Numerous studies have explored diverse methods to select ICDs in the NLP field [29, 45–48]. For example, [15] selects ICDs based on the embedding similarity between ICDs and test samples where the embeddings are extracted from an existing language encoder. Such a method is further developed by training an encoder specifically for selection [49–52, 50, 53].

Regarding the ordering of ICDs, researchers calculate diverse statistical-based metrics to measure the quality of ICD configurations, *e.g.*, the Minimal Description Length [29] and Global and Local Entropy [21]. Besides them, researchers focus more on discovering the statistical patterns of good prompts. For example, [30] uses perplexity to measure which prompts can better help LLMs perform a task. Furthermore, [31] unifies diverse statistics-based prompt selection methods [54, 55] from the perspective of mutual information and discover that mutual information or its variants can uncover certain statistical patterns of effective prompts. However, these statistical-based methods require to calculate the token probabilities, making them infeasible to address continuous image patches, thus can not be used in VL.

Besides these NLP studies, in VL, [24] and [25] explore diverse ICD configurations in IC and VQA, while only the heuristic-based methods are used for selecting ICDs and do not consider the ordering. In contrast, our Lever-LM can simultaneously learn how to select and reorder the samples and moreover, our Lever-LM is model-specific.

## 3 Lever Language Model

In this section, we introduce how to build Lever Language Model (Lever-LM) for configuring the ICD sequence to lever a given LVLM. First, we briefly introduce the formulations of ICL for Vision-Language (VL) tasks. Then we introduce the construction of the dataset used to train Lever-LM. After that, we show the architecture of Lever-LM and briefly discuss how to train Lever-LM and use it to configure ICD sequences.

**The Formulation of In-Context Learning (ICL).** Given a query input $\mathbf{x}'$, ICL predicts the corresponding output $\mathbf{y}'$ using a well-trained foundation model $\mathcal{M}$, conditioned on the concatenation of an in-context sequence $\mathcal{S}$ and this query. We denote a in-context sequence with $K$-shot ICDs $\hat{d}$ as $\mathcal{S}^K = \{\hat{d}_1, \hat{d}_2, ..., \hat{d}_K\}$. Then ICL can be formulated as:

$$\mathbf{y}' \leftarrow P_{\mathcal{M}}(\mathbf{y}' \mid \mathcal{S}^k, \mathbf{x}'), \tag{1}$$

where $P_{\mathcal{M}}$ denotes the predicted probability of $\mathcal{M}$ and "$\leftarrow$" represents the decoding strategy, *e.g.*, beam search. For each $\hat{d}$, it is selected from a supporting set $\mathcal{D}_S = \{d_1, ..., d_N\}$, where each sample $d_i = (\mathbf{x}_i, \mathbf{y}_i)$: $\mathbf{x}_i$ and $\mathbf{y}_i$ respectively denote the input and the corresponding label. It is noteworthy that in diverse VL tasks, $\mathbf{x}$ and $\mathbf{y}$ have different forms. For instance, in Image Captioning (IC), $\mathbf{x}$ is the image and $\mathbf{y}$ is the caption; and in Vision Question Answering (VQA), $\mathbf{x}$ contains the image and the question, while $\mathbf{y}$ is the answer.

---

[2]Since Flamingo does not open-source the model, we use an unofficial implementation, OpenFlamingo [34].

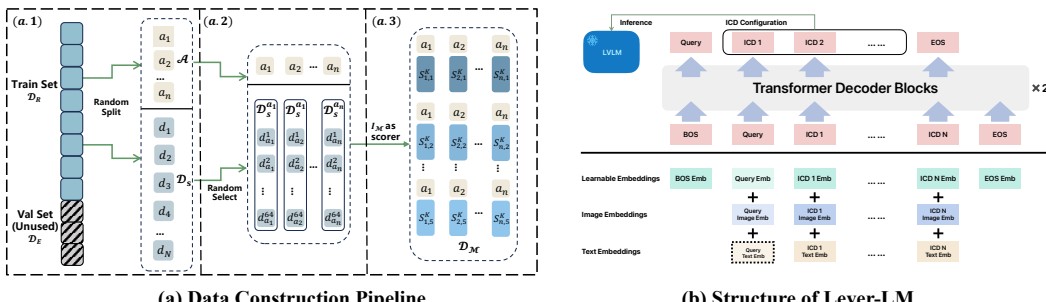

**(a) Data Construction Pipeline**      **(b) Structure of Lever-LM**

Figure 2: (a): The pipeline of constructing $\mathcal{D}_\mathcal{M}$. Darker color of $S_{i,j}^K$ indicates a higher score given by Eq. 2. (b): Top: Lever-LM is a two-layer Transformer. Bottom: Each input embeddings is the sum of the random initialized learnable embeddings, the image and text embeddings extracted by CLIP. The dotted block means that some tasks do not exist the text input, *e.g.*, IC.

**Constructing the Training Dataset.** To train Lever-LM for generating effective ICD sequences for a given LVLM $\mathcal{M}$, we should first construct a dataset $\mathcal{D}_\mathcal{M}$ containing high-quality ICD sequences for different query inputs. Simply, we use one VL dataset—COCO [56] from IC— to show how to construct $\mathcal{D}_\mathcal{M}$, which is shown in Fig. 2. Formally, given a dataset $\mathcal{D}$ which is already split into the training part $\mathcal{D}_R$ and the test part $\mathcal{D}_E$, we build $\mathcal{D}_\mathcal{M}$ only from $\mathcal{D}_R$. As Fig. 2 (a.1) shows, initially, we randomly select $n$ samples from $\mathcal{D}_R$ to form an anchor set $\mathcal{A}$. Then for each sample $\boldsymbol{a}_m = \{\boldsymbol{x}_m, \boldsymbol{y}_m\} \in \mathcal{A}$, we construct a $K$-shot in-context sequence $\mathcal{S}_m^K$ for it. Then $\mathcal{D}_\mathcal{M} = \{(\boldsymbol{a}_1, \mathcal{S}_1^K), (\boldsymbol{a}_2, \mathcal{S}_2^K), ..., (\boldsymbol{a}_M, \mathcal{S}_M^K)\}$ where each training sample contains an query $\boldsymbol{a}_m$ and the corresponding $K$-shot in-context sequence $\mathcal{S}_m^K$.

To avoid confusion, we remove the subscript $m$ in following texts. To construct $\mathcal{S}^K = \{\boldsymbol{d}_1, ..., \boldsymbol{d}_K\}$, we need to select $K$-shot samples from the supporting set $\mathcal{D}_S$, which is set to the complement set of $\mathcal{A}$ in $\mathcal{D}_R$: $\mathcal{D}_R \backslash \mathcal{A}$. Meantime, we also need to decide which samples should be selected in turn. To achieve this, given the anchor sample $\boldsymbol{a} = \{\boldsymbol{x}, \boldsymbol{y}\}$ and the partially constructed in-context sequence, *e.g.*, a $k-1$-shot $\mathcal{S}^{k-1}$, we need to know that after adding which sample $\boldsymbol{d} \in \mathcal{D}_S$, the ICL performance improvement can be maximized by applying the given LVLM $\mathcal{M}$:

$$\hat{\boldsymbol{d}}_k = \underset{\boldsymbol{d} \in \mathcal{D}_S}{\arg\max}\, I_\mathcal{M}(\{\boldsymbol{d}, \mathcal{S}^{k-1}\}, \boldsymbol{a}) - I_\mathcal{M}(\mathcal{S}^{k-1}, \boldsymbol{a}), \tag{2}$$

where $I_\mathcal{M}$ is one kind of ICL performance measurement related to $\mathcal{M}$. Note that Eq. (2) actually uses the greedy sampling method to select the samples every time, while we can use beam search here to further achieve a better solution. Additionally, to improve the diversity of the dataset, we will keep the top-$b$ highest-scoring ICD sequences $\{S_1^K, S_2^K, ..., S_b^K\}$ at the last iteration for each $a$, where $b$ is equal to the beam size. For example, when setting beam size to 5 as shown in Fig. 2 (a.3), we can get 5 diverse high-quality ICD sequences for an anchor sample.

Intuitively, for diverse tasks, we can use the corresponding "golden measurement" as $I_\mathcal{M}$, *e.g.*, setting it to CIDEr [57]/accuracy for IC/VQA. However, this strategy encounters two limitations. The first one is that for diverse VL tasks, we need diverse corresponding measurements, which is inconvenient. Second, some "golden measurements" may be impractical to deploy. For example, for IC, calculating CIDEr requires the LVLM to forward multiple times to sample an integral sentence, and then it costs expensive time burdens to construct the dataset. While for VQA, accuracy is a binary value (accuracy=1 when correct and 0 when wrong), then maybe lots of candidates in $\mathcal{D}_S$ will make the accuracy change from 0 to 1 and then it is hard to judge which one of them is the most suitable one.

To overcome these two limitations, we use a relatively general measurement as $I_\mathcal{M}$. Formally, since we have the ground-truth results $\boldsymbol{y}$ of the anchor sample, we can use the given LVLM $\mathcal{M}$ to measure the prediction confidence of $\boldsymbol{y}$ given the input $\boldsymbol{x}$ and the in-context sequence $\mathcal{S}^K$:

$$I_\mathcal{M}(\mathcal{S}^K, \boldsymbol{a}) = P_\mathcal{M}(\boldsymbol{y}|\mathcal{S}^K, \boldsymbol{x}) = \prod_t P_\mathcal{M}(y^{(t)}|\mathcal{S}^K, \boldsymbol{x}, y^{(1:t-1)}). \tag{3}$$

In VL tasks, the ground-truth label $\boldsymbol{y} = \{y^{(1)}, ..., y^{(T)}\}$ is a sequence, thus we can decompose the probability distribution into a series of productions. Then Eq. (2) selects a sample that can further maximize the prediction confidence given the query input and the current in-context sequence.

In implementation, $\mathcal{D}_S$ usually contains huge amounts of samples, *e.g.*, $\mathcal{D}_S$ in COCO [56] contains about $10^5$ samples. However, we need to calculate Eq. (2) for each $\boldsymbol{d} \in \mathcal{D}_S$ when selecting $\hat{\boldsymbol{d}}_k$ for each $\boldsymbol{a} \in \mathcal{A}$, which means the whole process of building $\boldsymbol{D}_{\mathcal{M}}$ is quite time-consuming. To alleviate the cost, as shown in Fig. 2 (a.2), for each specific $\boldsymbol{a}$, we narrow the set size by sampling a much smaller subset $\mathcal{D}_S^{\boldsymbol{a}}$, *e.g.*, containing 64 samples $\mathcal{D}_S^{\boldsymbol{a}} = \{d_{\boldsymbol{a}}^1, d_{\boldsymbol{a}}^2, ..., d_{\boldsymbol{a}}^{64}\}$, from $\mathcal{D}_S$ for selecting $\hat{\boldsymbol{d}}_k$. We use diverse sampling strategies to construct this subset, *e.g.*, retrieving some samples similar to $\boldsymbol{a}$, and implement exhaustive ablation studies to explore which strategies are useful in Section 4.3.

**Training Lever-LM.** After getting $\mathcal{D}_{\mathcal{M}}$, we use it to train Lever-LM, as Fig. 2(b) shows, it is a two-layer tiny Transformer [58]. The primary difference between Lever-LM and the traditional LM lies in the tokens of the vocabulary, whose tokens are the samples from the supporting set $\mathcal{D}_S$, *e.g.*, the first token corresponds to the first sample in $\mathcal{D}_S$. Then given the query sample, the ICDs can be selected one by one based on the token distribution produced by the trained Lever-LM, just as when composing a sentence, the words are selected one by one from the word vocabulary.

Besides the tokens from $\mathcal{D}_S$, three special tokens are added into the vocabulary to help configure the ICD sequence, which are [BOS], [EOS], and [QUERY], respectively representing the beginning of a sequence, the end of a sequence, and the query sample. Given a data sample $(\mathcal{S}^K = \{\boldsymbol{d}_1, ..., \boldsymbol{d}_K\}, \boldsymbol{x}')$ from $\mathcal{D}_{\mathcal{M}}$ where $\mathcal{S}^K$ is the ICD sequence and $\boldsymbol{x}'$ is the query input, we reformulate it into $\{[BOS], [QUERY] + \boldsymbol{x}', \boldsymbol{d}_1, ..., \boldsymbol{d}_K, [EOS]\}$ where $[QUERY] + \boldsymbol{x}'$ denotes to add two embeddings. This reformulated sequence is input into Lever-LM for training.

To train Lever-LM, we should embed the tokens of the vocabulary to get dense embeddings. Since each token contains both image and text, we use the vision encoder $F_I(\cdot)$ and the language encoder $F_T(\cdot)$ of CLIP [59] to embed the image and text, respectively. Meanwhile, we add each of these embeddings with a learnable part $\boldsymbol{r}_i$ that is randomly initialized. Then for the $i$-th token $\boldsymbol{d}_i = (I_i, T_i)$ in the vocabulary where $I_i/T_i$ are the corresponding image/ text, its token embedding is $\boldsymbol{e}_i$:

$$\boldsymbol{e}_i = F_I(I_i) + F_T(T_i) + \boldsymbol{r}_i. \tag{4}$$

Note that $T_i$ varies between IC and VQA tasks where it denotes caption in IC and question in VQA.

For test query $\boldsymbol{x}'$, we use the same vision and language encoders to embed it. For VQA, the image and question are embedded and summed, while for IC, only the image is embedded. Lastly, we use the cross-entropy loss for training as a standard LM that given the previously $k - 1$ tokens, we maximize the probability of the $k$-th ground-truth token.

**Configuring the ICD Sequence to Lever LVLM.** After training Lever-LM, we use it to configure the ICD sequence. Given a query sample $x'$, we initialize the input sequence as $\{[BOS], [QUERY] + e_{x'}\}$ and then generate the ICDs one by one, where $e_{x'}$ is the embedding of $x'$ computed by Eq. (4). After iteratively sampling $K$-shot ICDs, we can compose the corresponding in-context sequence $\mathcal{S}^K$ for $x'$ and then use Eq. (1) to implement the ICL.

## 4 Experiments

### 4.1 Datasets and implementation details

Our approach is evaluated on MS-COCO [56] for Image Captioning (IC) and VQAV2 [60] for Visual Question Answering (VQA). For each corresponding dataset, we use the train split to construct the $\mathcal{D}_{\mathcal{M}}$ and use the validation split to evaluate the performance of ICD configurations generated by Lever-LM. More details are given in Appendix A.

To get $\mathcal{D}_{\mathcal{M}}$, we select 5000 samples to get the anchor set $\mathcal{A}$. For each anchor sample, we randomly choose 64 samples to build the sub-supporting set $\mathcal{D}_S^{\boldsymbol{a}}$. The beam size for these processes is 5. To train Lever-LM, different strategies are employed for IC and VQA. In IC, the weight of CLIP model will be frozen, and an MLP adapter is introduced to its output. While, for VQA, the CLIP encoder remains trainable, and no adapter is appended. The training phase leverages the AdamW optimizer [61] and a cosine learning rate scheduler. We set the learning rate to $1 \times 10^{-4}$ and the batch size to 128. We train our Lever-LM for 20 epochs. To implement ICL, we use OpenFlamingoV2-9B [34] and IDEFICS-9B [14] as our LVLMs. We use beam search during inference where the beam size is set to 3. Besides, we set the maximum number of generated tokens as 20 in IC and 5 in VQA.

### 4.2 Results and Analyses

#### 4.2.1 Comparison Methods

We compare Lever-LM with 4 ICD selection strategies:

**Random Sample (RS)**: RS constructs $\mathcal{S}^k$ by randomly selecting and ordering $k$ ICDs from $\mathcal{D}_S$.

**Similarity-based Retrieval methods**: To date, only a few studies focus on configuring ICD sequence for solving VL tasks [24, 25], where both studies show that, despite their simplicity, similarity-based retrieval methods are effective for selecting ICDs. We therefore consider these strategies as current SOTA benchmarks to assess our effectiveness. [3]. They form $\mathcal{S}^k$ by computing the cosine similarity between the query input $\mathbf{x}'$ and ICDs in $\mathcal{D}_S$ where CLIP is used to extract features. We follow [25] to sort examples in ascending order by their similarity to the query input, so the rightmost demonstration is the closest example. Similarity-based methods contain three variants: (1). **Similarity-based Image-Image Retrieval (SIIR):** We select $k$ ICDs from $\mathcal{D}_S$ with highest image similarity to the query image. (2). **Similarity-based Text-Text Retrieval (STTR):** We select $k$ ICDs from $\mathcal{D}_S$ with highest text similarity to the query text. This method is only applicable to VQA where question is used as text and not infeasible for IC. (3). **Similarity-based Image-Text Retrieval (SITR):** We compute the similarity between query image and all text of $d_i \in \mathcal{D}_S$ and select ICDs whose texts have the top-$k$ similarities with the query image. For IC/VQA, we use caption/question for IC/VQA.

#### 4.2.2 Main Result

The results for various ICD selection strategies are shown in Table 1 for IC and VQA. For Lever-LM, it is trained by $\mathcal{D}_\mathcal{M}$ whose ICD length is set to 2. Due to increased inference time with more shots, we do not test the inference results for 5- and 7-shots. The table shows the length interpolation and extrapolation ability of Lever-LM. Interpolation refers to performance with ICDs shorter than those in the training set $\mathcal{D}_\mathcal{M}$, which contains only 2-shot ICDs, and is denoted as "Avg:1∼2". Extrapolation pertains to performance with ICDs longer than those in $\mathcal{D}_\mathcal{M}$, represented as "Avg:3∼8". The notation "Avg:1∼8" indicates overall performance across 1 to 8 shots. Future analysis will focus on comparing these averages to minimize potential bias across methods.

Overall, Lever-LM achieves the best performance on most cases compared to other methods on both LVLMs. Notably, Lever-LM excels in Avg:1∼2. Specifically, in VQA, Lever-LM surpasses the best performing SIIR method by 3.07 (48.75 vs. 45.68) and 0.57 (53.65 vs. 53.08) in accuracy on the IDEFICS and OpenFlamingo models, respectively. In IC, Lever-LM outperforms the best baseline, SIIR, by 6.03 (84.32 vs. 78.29) CIDEr on OpenFlamingo. Similarly, for IDEFICS, Lever-LM achieves a higher CIDEr of 3.2 (89.57 vs. 86.37) compared to the best baseline, RS.

Moreover, Lever-LM has remarkable extrapolation abilities. Regarding Avg:3∼8, Lever-LM maintains the top performance in both IC and VQA. Specially, on OpenFlamingo, Lever-LM outperforms SIIR with a 0.8 higher CIDEr in IC (96.52 vs. 95.72) and a 0.75 greater accuracy in VQA (52.59 vs. 51.84). Meanwhile, on IDEFICS, compared with RS, Lever-LM achieves a 2.93 higher CIDEr in IC (105.79 vs. 102.86) and 0.49 higher accuracy in VQA (54.84 vs. 54.35). These results indicate that *Lever-LM can effectively identify and utilize internal statistical patterns to compose longer, high-quality ICD sequences, even from a dataset comprising only two shots.*

When we delve deeper into the results in Table 1, we find that the relative performance of similarity-based methods and RS varies by LVLM and task. For example, for IC, SIIR outperforms RS ( Avg:1∼8: 89.91 vs. 88.48) when OpenFlamingo is used to implement ICL while SIIR significantly lags behind RS (Avg:1∼8: 88.19 vs. 97.36) when IDEFICS is used. Also, for VQA, STTR is comparable to RS (Avg:1∼8: 47.98 vs. 47.94) on OpenFlamingo while STTR is defeated by RS (Avg:1∼8: 49.75 vs. 53.54) when IDEFICS is used. These performance fluctuations demonstrates the instability of these heuristic-based methods. However, Lever-LM does not have such serious fluctuations where it outperforms both RS and similarity-based retrieval methods across various LVLMs and tasks on average. Such observations also suggest that *Lever-LM may capture the stable statistic patterns between ICDs.*

Besides the above-mentioned advantages, Fig. 3 shows that *Lever-LM is more robust to the Short-cut Inference brought by using similarity-based retrieval methods [24, 25].* For example, in (a) and

---

[3]Note that we use different experiment settings from [24] and more details are given in Appendix A.4

Table 1: Results of diverse ICL methods on IC and VQA, where "OF" and "IDE" denote Open-Flamingo and IDEFICS, respectively. Lever-LM is trained by $\mathcal{D}_\mathcal{M}$ whose ICD length is set to 2.

| | | Interpolation | | | Extrapolation | | | | | Avg:1~8 |
|---|---|---|---|---|---|---|---|---|---|---|
| | | Shot 1 | Shot 2 | Avg:1~2 | Shot 3 | Shot 4 | Shot 6 | Shot 8 | Avg:3~8 | |
| OF IC | RS | 73.32 | 82.95 | 78.14 | 87.72 | 93.65 | 95.81 | 97.42 | 93.65 | 88.48 |
| | SITR | 66.05 | 77.69 | 71.87 | 83.46 | 85.05 | 89.84 | 93.57 | 87.98 | 82.61 |
| | SIIR | 71.71 | 84.87 | 78.29 | 90.83 | 93.22 | **97.80** | **101.01** | 95.72 | 89.91 |
| | Lever-LM | **80.02** | **88.63** | **84.32** | **93.41** | **96.06** | 97.26 | 99.35 | **96.52** | **92.45** |
| OF VQA | RS | 41.97 | 45.92 | 43.95 | 48.17 | 48.95 | 51.18 | 51.44 | 49.94 | 47.94 |
| | SITR | 40.17 | 43.58 | 41.88 | 46.03 | 47.5 | 49.72 | 50.75 | 48.50 | 46.29 |
| | SIIR | 43.31 | 47.46 | 45.39 | 49.85 | 50.68 | 53.23 | **53.58** | 51.84 | 49.69 |
| | STTR | 44.6 | 46.75 | 45.68 | 47.92 | 49.05 | 50.06 | 49.47 | 49.13 | 47.98 |
| | Lever-LM | **46.66** | **50.83** | **48.75** | **51.91** | **52.15** | **53.29** | 53.01 | **52.59** | **51.31** |
| IDE IC | RS | 76.44 | 96.31 | 86.37 | 100.80 | 101.82 | 103.64 | 105.18 | 102.86 | 97.36 |
| | SITR | 62.08 | 75.50 | 68.79 | 82.57 | 86.64 | 90.34 | 92.88 | 88.11 | 81.67 |
| | SIIR | 66.61 | 83.31 | 74.96 | 89.43 | 93.02 | 97.07 | 99.70 | 94.81 | 88.19 |
| | Lever-LM | **78.70** | **100.45** | **89.57** | **104.52** | **104.86** | **106.44** | **107.33** | **105.79** | **100.38** |
| IDE VQA | RS | 52.40 | 53.21 | 52.81 | 53.47 | 53.70 | 54.00 | 54.48 | 53.91 | 53.54 |
| | SITR | 51.52 | 51.72 | 51.62 | 52.55 | 52.59 | 52.83 | 50.49 | 52.12 | 51.95 |
| | SIIR | 52.87 | 53.28 | 53.08 | 53.77 | 53.92 | 54.81 | 54.88 | 54.35 | 53.92 |
| | STTR | 48.13 | 49.32 | 48.73 | 49.43 | 50.11 | 50.58 | 50.93 | 50.26 | 49.75 |
| | Lever-LM | **53.31** | **53.98** | **53.65** | **54.39** | **54.58** | **55.09** | **55.3** | **54.84** | **54.44** |

(b), all the ICD questions are yes-or-no type, which causes LVLMs to output "no" even when faced with a "What" type question of the query sample. For IC in (c) and (d), SITR incorrectly outputs "London" even though the location is not explicitly indicated in the query image, while SIIR even leads LVLMs to directly copy the text from the ICD. Conversely, Lever-LM generates more diverse ICD configurations, thereby preventing misleading inferences.

## 4.3 Ablation Studies

We use ablation studies to explore the effects of diverse settings on our approach, including (1) diverse $\mathcal{D}_\mathcal{M}$ configurations; (2) diverse scorers $I_\mathcal{M}$ in Eq. (3); (3) diverse LM structures; (4) $\mathcal{D}_\mathcal{M}$ with 4-shot ICDs; and (5) randomly ordering the ICD sequences generated by Lever-LM.

**Diverse $\mathcal{D}_\mathcal{M}$ Configurations.** We generate 2-shot $\mathcal{D}_\mathcal{M}$ by different settings to investigate the corresponding effects. Three factors are ablated: beam size $b$; the number $n$ of samples in $\mathcal{A}$; and the sampling method of $\mathcal{D}_S^a$, including 3 methods: **Random**: Selecting randomly from $\mathcal{D}_S$; **Similar Text (Sim-T)**: Selecting the highest textual similarity sample with anchor sample $a$ from $\mathcal{D}_S$; **Similar Image (Sim-I)**: Selecting the highest visual similarity sample with anchor sample $a$ from $\mathcal{D}_S$.

Table 2 (3) $\sim$ (11) shows the results for different $\mathcal{D}_\mathcal{M}$ configurations on IC and VQA. We find that Lever-LM can consistently improve the performance compared with the baseline RS [4] in Avg:1~2. As for length extrapolation capability, only *Sim-T* gets a lower score than RS. These comparisons confirm the robustness of our method.

Table 2 (3) $\sim$ (5) show that appropriately increasing the beam size $b$ can improve performance. Specifically, as $b$ increases from 1 to 5, in Avg:1~8, CIDEr/accuracy increases by 2.79/1.73 for IC/VQA. This suggests that a diverse $\mathcal{D}_\mathcal{M}$ encompasses a broader range of high-quality ICD configurations, which can help train a better Lever-LM. However, an excessively large $b$ can negatively impact performance. For instance, in VQA, the accuracy of $b = 10$ decays 0.12 than $b = 5$ in Avg:1~8. We hypothesize this drop in performance is due to the introduction of lower-scoring ICD sequences with a large beam size, potentially misleading Lever-LM during training.

Table 2 (6) $\sim$ (8) show that using more anchor samples can improve the interpolation performance in both IC and VQA, *e.g.*, when $n$ increases from 1000 to 5000, the Avg:1~2 CIDEr/accuracy of IC/VQA increases from 83.94/45.39 to 84.32/ 48.75. However, we find that on IC, although the interpolation performance increases when $n$ changes from 3000 to 5000, the extrapolation performance decays, *e.g.*, Avg:3~8 decreases from 97.60 to 96.52. One possible reason is that using

---

[4]The RS means Random Sample ICD retrieval method which is mentioned in Section 4.2.1.

Table 2: Results of diverse ablation studies on IC and VQA.

| | | IC | | | VQA | | |
| --- | --- | --- | --- | --- | --- | --- | --- |
| | | Avg:1~2 | Avg:3~8 | Avg:1~8 | Avg:1~2 | Avg:3~8 | Avg:1~8 |
| (1) | RS | 78.14 | 93.65 | 88.48 | 43.95 | 49.94 | 47.94 |
| (2) | Lever-LM | 84.32 | 96.52 | 92.45 | 48.75 | 52.59 | 51.31 |
| (3) | $b = 1$ | 79.91 | 94.53 | 89.66 | 46.47 | 51.13 | 49.58 |
| (4) | $b = 5$ | 84.32 | 96.52 | 92.45 | **48.75** | **52.59** | **51.31** |
| (5) | $b = 10$ | **84.96** | **97.12** | **93.06** | 48.58 | 52.49 | 51.19 |
| (6) | $n = 1000$ | 83.94 | 96.74 | 92.48 | 45.39 | 50.44 | 48.76 |
| (7) | $n = 3000$ | 84.13 | **97.60** | **93.11** | 47.56 | 51.11 | 49.93 |
| (8) | $n = 5000$ | **84.32** | 96.52 | 92.45 | **48.75** | **52.59** | **51.31** |
| (9) | Sim-I | 81.96 | 96.11 | 91.40 | 47.10 | 51.79 | 50.23 |
| (10) | Sim-T | 81.22 | 87.66 | 85.52 | 45.38 | 49.55 | 48.16 |
| (11) | Random | **84.32** | **96.52** | **92.45** | **48.75** | **52.59** | **51.31** |
| (12) | CIDEr Scorer | 87.93 | 93.52 | 91.65 | - | - | - |
| (13) | Lever-LM LSTM | 83.93 | 96.21 | 92.12 | 46.60 | 50.68 | 49.32 |
| (14) | Golden-1 | 81.78 | 97.44 | 92.22 | 47.78 | 52.51 | 50.93 |
| (15) | Golden-2 | 91.20 | 99.63 | 96.82 | 45.32 | 49.05 | 47.80 |

Table 3: Results of Lever-LM with 4-shot $\mathcal{D}_{\mathcal{M}}$ on IC and VQA.

| Model | IC | | | VQA | | |
| --- | --- | --- | --- | --- | --- | --- |
| | Avg:1~4 | Avg:6~8 | Avg:1~8 | Avg:1~4 | Avg:6~8 | Avg:1~8 |
| RS | 84.41 | 96.62 | 88.48 | 46.25 | 51.31 | 47.94 |
| SITR | 78.06 | 91.71 | 82.61 | 44.32 | 50.24 | 46.29 |
| SIIR | 85.16 | **99.40** | 89.91 | 47.83 | **53.41** | 49.69 |
| STTR | - | - | - | 47.08 | 49.77 | 47.98 |
| Lever-LM(4-shot $\mathcal{D}_{\mathcal{M}}$) | **87.35** | 97.96 | **90.88** | **48.56** | 52.68 | **49.93** |
| Lever-LM(2-shot $\mathcal{D}_{\mathcal{M}}$) | 89.53 | 98.30 | 92.45 | 50.39 | 53.15 | 51.31 |

Eq. (3) to build $D_{\mathcal{M}}$ may introduce certain in-domain bias which is beneficial for interpolation while detrimental for extrapolation on IC.

For different sample methods of constructing the $\mathcal{D}_S^a$ in table 2 (9) ~ (11), we find Random is the best in both IC and VQA. We suppose this is because selecting similar ICDs with the anchor sample from $\mathcal{D}_S$ will damage the diversity. Previous study [62] in NLP validates that the diversity of the ICD sequences will also help improve the performance of LLMs.

**Diverse scorers $I_{\mathcal{M}}$ for evaluating ICD sequences.** To evaluate the quality of ICD configurations, we can use task-specific metrics as $I_{\mathcal{M}}$ to build $\mathcal{D}_{\mathcal{M}}$, such as CIDEr in IC. Table 2 (2) and (12) compare the results between using prediction confidence Eq. 3 (2) and CIDEr (12) as $I_{\mathcal{M}}$. We find that using CIDEr achieves 3.61 higher than Confidence in Avg:1~2, suggesting that CIDEr can assign a more accurate and reasonable score for ICD configurations. However, the length extrapolation capability decreases obviously, which is 3.0 lower than Confidence in Avg:3~8, validating the robustness of Confidence scorer. Moreover, it will cost more time to construct $\mathcal{D}_{\mathcal{M}}$ by task-specific metric is used, *e.g.*, CIDEr costs approximately 10 times of Confidence when constructing $\mathcal{D}_{\mathcal{M}}$.

**Diverse LM Structures.** Table 2 (13) shows the results of using LSTM [63] as Lever-LM, we find that this still achieves excellent performance. For example, in IC, the overall performance improves by 3.64 (92.12 vs. 88.48) compared to the RS baseline, while in VQA, it is improved by 1.38 (49.32 vs. 47.94). However, due to the weak representation learning capability of LSTM, its performance is lower than Transformer,*e.g.*, the scores decrease by 0.33/1.99 in IC/VQA, respectively. Overall, these results suggest that effective ICD configurations contain internal statistic patterns which can be captured by different temporal learner.

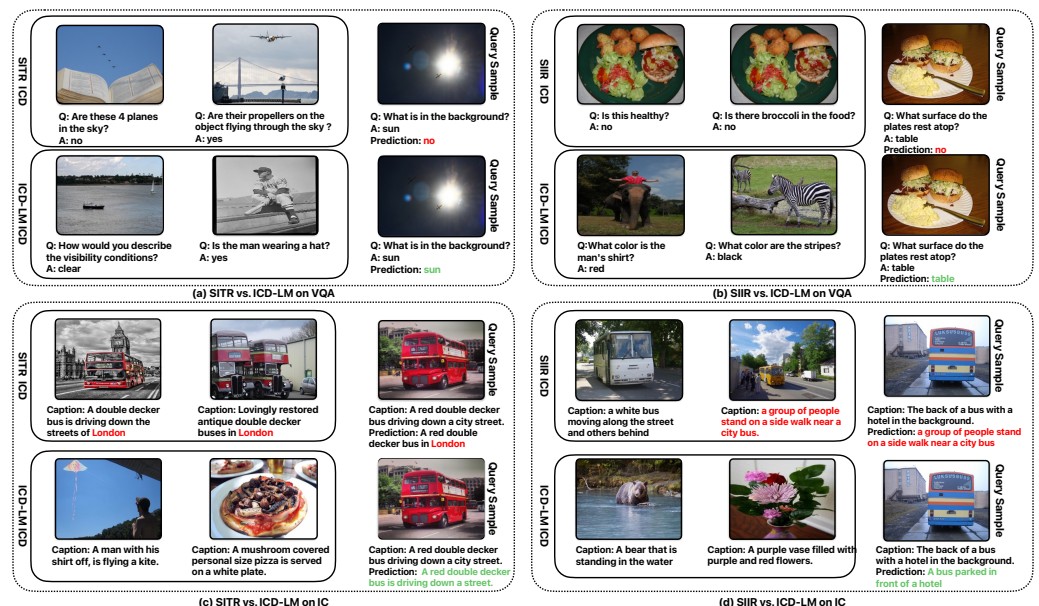

Figure 3: Visualizations of diverse ICDs configurations, where the first and the last ICDs are given due to space limitation. We can find that Lever-LM use more diverse ICDs and thus not lead to short-cut inference.

**Golden ICD Sequence.** In experiments, we find that using a high learning rate and not freezing the CLIP model may make Lever-LM converge to a specific solution that for any query input, the ICD configuration is fixed while still return good ICL performance. For example, in IC,

Table 4: Results of Random Order of Lever-LM generated ICDs.

|  | 2-Shot $\mathcal{D}_{\mathcal{M}}$ | | 4-Shot $\mathcal{D}_{\mathcal{M}}$ | |
|---|---|---|---|---|
|  | VQA | IC | VQA | IC |
| Original | **50.83** | **88.63** | **51.12** | **85.97** |
| Random Order | 50.42 | 88.56 | 50.63 | 85.77 |

the best version, Golden-2, can outperform the non-Fixed case (Table 2 (2)) by 4.37 points in Avg:1∼8. Such improvement suggest that if we do not have enough computation burdens to configure diverse ICD sequence for each query, we can preserve one Golden ICD Sequence for the latter usage. However, we also find that the performance of Golden ICD Sequence fluctuates significantly, *e.g.*, Golden-2 is poorer than RS in VQA in Avg:1∼8. This also points out a new direction to study how to get more stable Golden ICD Sequence.

**Longer Few-shot $\mathcal{D}_{\mathcal{M}}$.** We further explore the performance of Lever-LM using a 4-shot $\mathcal{D}_{\mathcal{M}}$, as presented in Table 3. It is evident that Lever-LM continues to outperform other retrieval-based methods in Avg:1∼8 metric. However, we observe a notable performance reduction when Lever-LM is trained with the 4-shot $\mathcal{D}_{\mathcal{M}}$ compared to the 2-shot $\mathcal{D}_{\mathcal{M}}$. Specifically, for IC, there is a performance decrease of approximately 1.57 in the Avg:1∼8 metric when using the 4-shot $\mathcal{D}_{\mathcal{M}}$ to train Lever-LM. One possible reason is that when constructing $\mathcal{D}_{\mathcal{M}}$, some approximation operations are applied where sub-optimal ICD sequences are got. Then parts of statistic patterns may be salient is $\mathcal{D}_{\mathcal{M}}$ that are more easily captured by Lever-LM where using longer ICD sequences may further encourage Lever-LM to capture such patterns, and thus causing less effective ICD sequences than the shorter $\mathcal{D}_{\mathcal{M}}$. This points out a new direction to study how to build $\mathcal{D}_{\mathcal{M}}$ with longer and more robust ICD sequences for better training Lever-LM.

**Random Order ICD sequence.** To validate that whether Lever-LM captures effective ICD orders, we randomly rearrange the ICD sequences generated by Lever-LM trained with 2-shot and 4-shot $\mathcal{D}_{\mathcal{M}}$ and then evaluate the performance the 2-shot and 4-shot ICD configurations, respectively, in Table 4. It is evident that the original order of ICDs generated by Lever-LM attains the highest score in both VQA and IC, validating that Lever-LM can learn how to order the ICDs.

## 5    Conclusion

After observing that configuring an ICD sequence is a mirror process of composing a sentence, we assume effective ICDs may contain statistic patterns that can be captured by temporal learner. Then we use a tiny LM named as Lever-LM to capture such patterns for configuring ICDs to lever LVLMs. To achieve this, we construct a dataset containing effective ICD sequences to train this Lever-LM. After training, we validate the effectiveness of Lever-LM by comparing it with similarity-based retrieval methods and find that Lever-LM can capture the statistic patterns between ICDs. Extensive ablations are deployed to discover which factors and why they will affect the results, which also pointing out a few future research directions.

## Acknowledgement

This work is supported by the National Science Foundation of China (62206048), the Natural Science Foundation of Jiangsu Province (BK20220819), the Young Elite Scientists Sponsorship Program of Jiangsu Association for Science and Technology (Tj-2022-027), and the Fundamental Research Funds for the Central Universities (2242024k30035). This research work is also supported by the Big Data Computing Center of Southeast University.

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

# A Implementation Details.

## A.1 Lever-LM Hyperparameters

We provide additional implementation details for each experiment in Table 5. For all experiments, the batch size is set to 64, the warmup steps to $5\%$ of total training steps, the scheduler as a cosine scheduler, and the optimizer as AdamW [61]. All experiments are deployed on an RTX 3090. All training processes are carried out with mixed precision and 2 RTX3090 GPUs. While for LVLM ICL experiments, we use BF16 mode.

Table 5: Different settings of Lever-LM experiments, where the $n$ is the number of anchor samples in $\mathcal{A}$, $b$ is the beam size, and $l$ is the length of ICD configurations.

| | | Training Parameters | | | | | $\mathcal{D}_{\mathcal{M}}$ Parameters | | | Avg:1~8 |
| | | lr | weight decay | epoch | Freeze | Adapter | $n$ | $b$ | $l$ | |
|---|---|---|---|---|---|---|---|---|---|---|
| IC | 2-shot $\mathcal{D}_{\mathcal{M}}$ (OpenFlamingo) | $1.0 \times 10^{-4}$ | $1.0 \times 10^{-3}$ | 20 | ✓ | ✓ | 5000 | 5 | 2 | 92.45 |
| | 2-shot $\mathcal{D}_{\mathcal{M}}$ (IDEFICS) | $1.0 \times 10^{-4}$ | $1.0 \times 10^{-3}$ | 20 | ✓ | ✓ | 5000 | 5 | 2 | 100.38 |
| | 4-shot $\mathcal{D}_{\mathcal{M}}$ | $1.0 \times 10^{-4}$ | $1.0 \times 10^{-3}$ | 20 | ✓ | ✓ | 10000 | 5 | 4 | 90.88 |
| | Lever-LM(LSTM) | $1.0 \times 10^{-3}$ | $1.0 \times 10^{-3}$ | 20 | ✓ | ✓ | 5000 | 5 | 2 | 92.12 |
| | CIDEr Scorer | $1.0 \times 10^{-4}$ | $1.0 \times 10^{-3}$ | 20 | ✓ | ✓ | 5000 | 5 | 2 | 91.65 |
| | Fixed Set-1 | $5.0 \times 10^{-3}$ | $1.0 \times 10^{-3}$ | 10 | ✗ | ✗ | 5000 | 5 | 2 | 92.22 |
| | Fixed Set-2 | $1.0 \times 10^{-3}$ | $1.0 \times 10^{-3}$ | 20 | ✗ | ✗ | 5000 | 5 | 2 | 96.82 |
| VQA | 2-shot $\mathcal{D}_{\mathcal{M}}$ (OpenFlamingo) | $1.0 \times 10^{-4}$ | $1.0 \times 10^{-3}$ | 20 | ✗ | ✗ | 5000 | 5 | 2 | 51.31 |
| | 2-shot $\mathcal{D}_{\mathcal{M}}$ (IDEFICS) | $1.0 \times 10^{-4}$ | $1.0 \times 10^{-3}$ | 20 | ✗ | ✗ | 5000 | 5 | 2 | 54.44 |
| | 4-shot $\mathcal{D}_{\mathcal{M}}$ | $1.0 \times 10^{-4}$ | $1.0 \times 10^{-1}$ | 20 | ✗ | ✗ | 10000 | 5 | 4 | 49.61 |
| | Lever-LM(LSTM) | $1.0 \times 10^{-5}$ | $1.0 \times 10^{-3}$ | 30 | ✗ | ✗ | 5000 | 5 | 2 | 49.32 |
| | Fixed Set-1 | $1.0 \times 10^{-3}$ | $1.0 \times 10^{-3}$ | 20 | ✗ | ✗ | 5000 | 5 | 2 | 50.93 |
| | Fixed Set-2 | $1.0 \times 10^{-3}$ | $1.0 \times 10^{-3}$ | 10 | ✗ | ✗ | 5000 | 5 | 2 | 47.80 |

## A.2 Datasets

**MS-COCO [56]** is widely used in IC, which is divided into $118,287$ training, $5,000$ validation, and $5,000$ test image-caption pairs. Notably, each training image is associated with five distinct human-annotated captions.

**VQAV2 [60]** emphasizes open-ended VQA tasks, which encompasses $4,437,570$ question-answer pairs in its training split, supplemented by an additional $2,143,540$ pairs in the validation split.

## A.3 Prompt Template

For different LVLMs and tasks, the input format also varies. We followed the prompt templates provided by OpenFlamingo and IDEFICS for our experiments. We show the prompt templates in table 6. In the VQA tests of IDEFICS, an additional instruction needs to be added at the beginning of the input. Therefore, our final input format of k-shot is: [instruction] + [ICD1 prompt] + [ICD2 prompt] + ... [ICDk prompt] + [Query prompt].

## A.4 Similarity-based retrieval method

In our study, we adopt the methodology proposed by [24] as a baseline, yet we incorporate several distinct experimental settings. Firstly, unlike their use of OpenFlamingoV1 [34] which employs LLaMA [5] as the underlying LLM, we utilize OpenFlamingoV2, based on the MPT LLM [37]. This version of OpenFlamingo leverages a more extensive dataset and a more advanced LLM, resulting in enhanced robustness and improved ICL capabilities.

Secondly, while they focus exclusively on ICL strategies in IC, which lacks textual input, our research extends to VQA. VQA involves textual queries; hence, we have adapted the STTR retrieval method for sourcing samples with similar textual prompts as ICDs, drawing inspiration from their approach.

Lastly, in the context of Image Captioning, our experiment utilizes the MSCOCO 2017 dataset, in contrast to their employment of the Karpathy split [64] of the MSCOCO 2014 dataset.

Table 6: Prompt Formats for IC and VQA tasks with placeholders.

| LVLM | Task | Prompt Template | Instruction |
|------|------|-----------------|-------------|
| IDEFICS | IC | `Caption:<X>` | - |
| | VQA | `Question:<Q> Short answer:<A>` | provide an answer to the question. Use the image to answer. |
| OpenFlamingo | IC | `Output:<X>` | - |
| | VQA | `Question:<Q> Short answer:<A>` | - |

Table 7: Results of diverse $\mathcal{D}_{\mathcal{M}}$ configurations on IC and VQA.

| | | Interpolation | | | Extrapolation | | | | | Avg:1~8 |
|---|---|---|---|---|---|---|---|---|---|---|
| | | Shot 1 | Shot 2 | Avg:1~2 | Shot 3 | Shot 4 | Shot 6 | Shot 8 | Avg:3~8 | |
| IC | RS | 73.32 | 82.95 | 78.14 | 87.72 | 93.65 | 95.81 | 97.42 | 93.65 | 88.48 |
| | $b=1$ | 75.67 | 84.15 | 79.91 | 90.10 | 92.93 | 95.92 | 99.16 | 94.53 | 89.66 |
| | $b=5$ | 80.02 | 88.63 | 84.32 | 93.41 | 96.06 | 97.26 | **99.35** | 96.52 | 92.45 |
| | $b=10$ | **80.64** | **89.27** | **84.96** | **94.47** | **96.26** | **98.58** | 99.15 | **97.12** | **93.06** |
| | $n=1000$ | 79.55 | 88.34 | 83.94 | 91.16 | 96.25 | 99.70 | 99.87 | 96.74 | 92.48 |
| | $n=3000$ | 79.30 | **88.96** | 84.13 | 93.28 | **97.66** | **99.34** | 100.15 | **97.60** | **93.11** |
| | $n=5000$ | **80.02** | 88.63 | 84.32 | 93.41 | 96.06 | 97.26 | 99.35 | 96.52 | 92.45 |
| | Sim-I | 75.32 | 88.61 | 81.96 | 93.28 | 95.28 | **97.71** | 98.19 | 96.11 | 91.40 |
| | Sim-T | 75.78 | 86.66 | 81.22 | 84.39 | 84.27 | 88.47 | 93.53 | 87.66 | 85.52 |
| | Random | **80.02** | **88.63** | **84.32** | **93.41** | **96.06** | 97.26 | **99.35** | **96.52** | **92.45** |
| VQA | RS | 41.97 | 45.92 | 43.95 | 48.17 | 48.95 | 51.18 | 51.44 | 49.94 | 47.94 |
| | $b=1$ | 44.56 | 48.38 | 46.47 | 49.76 | 49.95 | 52.27 | 52.54 | 51.13 | 49.58 |
| | $b=5$ | 46.66 | **50.83** | **48.75** | **51.91** | 52.15 | **53.29** | 53.01 | **52.59** | **51.31** |
| | $b=10$ | **46.89** | 50.27 | 48.58 | 51.54 | **52.17** | 53.00 | **53.26** | 52.49 | 51.19 |
| | $n=1000$ | 44.66 | 46.12 | 45.39 | 47.96 | 50.15 | 52.09 | 51.56 | 50.44 | 48.76 |
| | $n=3000$ | 46.04 | 49.08 | 47.56 | 50.42 | 50.88 | 51.54 | 51.59 | 51.11 | 49.93 |
| | $n=5000$ | **46.66** | **50.83** | **48.75** | **51.91** | 52.15 | **53.29** | 53.01 | **52.59** | **51.31** |
| | Sim-I | 45.00 | 49.19 | 47.10 | 50.49 | 51.38 | 52.64 | 52.66 | 51.79 | 50.23 |
| | Sim-T | 44.30 | 46.45 | 45.38 | 48.36 | 48.95 | 50.46 | 50.44 | 49.55 | 48.16 |
| | Random | **46.66** | **50.83** | **48.75** | **51.91** | **52.15** | **53.29** | **53.01** | **52.59** | **51.31** |

Table 8: CIDEr of diverse scorers on IC.

| | Interpolation | | | Extrapolation | | | | | Avg:1~8 |
|---|---|---|---|---|---|---|---|---|---|
| | Shot 1 | Shot 2 | Avg:1~2 | Shot 3 | Shot 4 | Shot 6 | Shot 8 | Avg:3~8 | |
| RS | 73.32 | 82.95 | 78.14 | 87.72 | 93.65 | 95.81 | 97.42 | 93.65 | 88.48 |
| Confidence | 80.02 | 88.63 | 84.32 | **93.41** | **96.06** | **97.26** | **99.35** | **96.52** | **92.45** |
| CIDEr | **84.86** | **90.99** | **87.93** | 92.53 | 94.25 | 94.98 | 92.31 | 93.52 | 91.65 |

Table 9: Results of diverse LM structures on IC and VQA.

| | | Interpolation | | | Extrapolation | | | | | Avg:1~8 |
|---|---|---|---|---|---|---|---|---|---|---|
| | | Shot 1 | Shot 2 | Avg:1~2 | Shot 3 | Shot 4 | Shot 6 | Shot 8 | Avg:3~8 | |
| IC | RS | 73.32 | 82.95 | 78.14 | 87.72 | 93.65 | 95.81 | 97.42 | 93.65 | 88.48 |
| | Lever-LM(LSTM) | 79.73 | 88.14 | 83.93 | 92.36 | **96.89** | 96.99 | 98.63 | 96.21 | 92.12 |
| | Lever-LM(Transformer) | **80.02** | **88.63** | **84.32** | **93.41** | 96.06 | **97.26** | **99.35** | **96.52** | **92.45** |
| VQA | RS | 41.97 | 45.92 | 43.95 | 48.17 | 48.95 | 51.18 | 51.44 | 49.94 | 47.94 |
| | Lever-LM(LSTM) | 44.64 | 48.55 | 46.60 | 49.77 | 50.51 | 50.71 | 51.72 | 50.68 | 49.32 |
| | Lever-LM(Transformer) | **46.66** | **50.83** | **48.75** | **51.91** | **52.15** | **53.29** | **53.01** | **52.59** | **51.31** |

# B    Detail Experiments Result

In this section, we present detailed experimental results and ablation studies on OpenFlamingo and IDEFICS models. The results include CIDEr scores, CHAIR metrics, fine-grained analyses, model size comparisons, inference time comparisons, and CLIPScore evaluations.

## B.1    Detail Ablation Studies Results On OpenFlamingo

In Table 7, Table 8, Table 9, Table 8, Table 10 and Table 11, we present all the detail ablation studies results on OpenFlamingo.

## B.2    CIDEr Results of Fixed Random ICD Sequence on Image Captioning

To evaluate the effectiveness of Golden-Set, we compare the fixed set learned by Lever-LM with the one constructed by random selection. Specifically, we use OpenFlamingo to randomly select 3 sets of k-shot ICD sequences with different seeds for the image captioning task and then conduct ICL

Table 10: Results of Lever-LM with 4-shot $\mathcal{D}_{\mathcal{M}}$ on IC and VQA.

| | | Interpolation | | | | | Extrapolation | | | Avg:1~8 |
|---|---|---|---|---|---|---|---|---|---|---|
| | | Shot 1 | Shot 2 | Shot 3 | Shot 4 | Avg:1~4 | Shot 6 | Shot 8 | Avg:6~8 | |
| IC | RS | 73.32 | 82.95 | 87.72 | 93.65 | 84.41 | 95.81 | 97.42 | 96.62 | 88.48 |
| | SITR | 66.05 | 77.69 | 83.46 | 85.05 | 78.06 | 89.84 | 93.57 | 91.71 | 82.61 |
| | SIIR | 71.71 | 84.87 | 90.83 | 93.22 | 85.16 | **97.80** | **101.01** | **99.40** | 89.91 |
| | Lever-LM(4-shot $\mathcal{D}_{\mathcal{M}}$) | **75.73** | **85.97** | **91.28** | **96.41** | **87.35** | 97.56 | 98.35 | 97.96 | **90.88** |
| | Lever-LM(2-shot $\mathcal{D}_{\mathcal{M}}$) | 80.02 | 88.63 | 93.41 | 96.06 | 89.53 | 97.26 | 99.35 | 98.30 | 92.45 |
| VQA | RS | 41.97 | 45.92 | 48.17 | 48.95 | 46.25 | 51.18 | 51.44 | 51.31 | 47.94 |
| | SITR | 40.17 | 43.58 | 46.03 | 47.5 | 44.32 | 49.72 | 50.75 | 50.24 | 46.29 |
| | SIIR | 43.31 | 47.46 | 49.85 | 50.68 | 47.83 | **53.23** | **53.58** | **53.41** | 49.69 |
| | STTR | 44.6 | 46.75 | 47.92 | 49.05 | 47.08 | 50.06 | 49.47 | 49.77 | 47.98 |
| | Lever-LM(4-shot $\mathcal{D}_{\mathcal{M}}$) | **44.55** | **48.05** | **50.65** | **50.98** | **48.56** | 52.43 | 52.92 | 52.68 | **49.93** |
| | Lever-LM(2-shot $\mathcal{D}_{\mathcal{M}}$) | 46.66 | 50.83 | 51.91 | 52.15 | 50.39 | 53.29 | 53.01 | 53.15 | 51.31 |

Table 11: Results of Fixed Set Lever-LM on IC and VQA.

| | | Interpolation | | | Extrapolation | | | | | Avg:1~8 |
|---|---|---|---|---|---|---|---|---|---|---|
| | | Shot 1 | Shot 2 | Avg:1~2 | Shot 3 | Shot 4 | Shot 6 | Shot 8 | Avg:3~8 | |
| IC | RS | 73.32 | 82.95 | 78.14 | 87.72 | 93.65 | 95.81 | 97.42 | 93.65 | 88.48 |
| | SIIR | 71.71 | 84.87 | 78.29 | 90.83 | 93.22 | 97.80 | 101.01 | 95.72 | 89.91 |
| | non Fixed Set | 80.02 | 88.63 | 84.32 | 93.41 | 96.06 | 97.26 | 99.35 | 96.52 | 92.45 |
| | Fixed Set-1 | 75.41 | 88.14 | 81.78 | 92.48 | 94.20 | 99.22 | **103.86** | 97.44 | 92.22 |
| | Fixed Set-2 | **86.37** | **96.03** | **91.20** | 89.80 | 98.81 | **105.39** | 103.51 | **99.63** | **96.82** |
| VQA | RS | 41.97 | 45.92 | 43.95 | 48.17 | 48.95 | 51.18 | 51.44 | 49.94 | 47.94 |
| | SIIR | 43.31 | 47.46 | 45.39 | 49.85 | 50.68 | 53.23 | 53.58 | 51.84 | 49.69 |
| | non Fixed Set | **46.66** | 50.83 | **48.75** | **51.91** | **52.15** | 53.29 | 53.01 | **52.59** | **51.31** |
| | Fixed Set-1 | 44.24 | **51.32** | 47.78 | 50.20 | 51.60 | **54.39** | **53.83** | 52.51 | 50.93 |
| | Fixed Set-2 | 44.24 | 46.40 | 45.32 | 48.38 | 48.41 | 49.16 | 50.23 | 49.05 | 47.80 |

Table 12: CIDEr results of a fixed random ICD sequences and Golden-Set on Image Captioning with OpenFlamingo.

| Method | Avg:1~2 | Avg:3~8 | Avg:1~8 |
|---|---|---|---|
| Random Fix-1 | 68.95 | 84.62 | 79.40 |
| Random Fix-2 | 64.80 | 82.15 | 76.36 |
| Random Fix-3 | 68.84 | 86.61 | 80.68 |
| **Golden-Set** | **91.20** | **99.63** | **96.82** |

Table 13: Fine-grained analysis of diverse ICL methods on VQAv2 with IDEFICSv1.

| Type | Method | Avg:1~2 | Avg:4~8 | Avg:1~8 |
|---|---|---|---|---|
| Yes/No | Lever-LM | **0.5623** | **0.5939** | **0.5833** |
| | RS | 0.5402 | 0.5795 | 0.5664 |
| | SIIR | 0.5593 | 0.5853 | 0.5799 |
| | SITR | 0.5229 | 0.5652 | 0.5511 |
| Other | Lever-LM | **0.3192** | **0.3574** | **0.3446** |
| | RS | 0.2852 | 0.3267 | 0.3129 |
| | SIIR | 0.2917 | 0.3349 | 0.3205 |
| | SITR | 0.2643 | 0.3049 | 0.2914 |
| Counting | Lever-LM | **0.2782** | **0.3079** | **0.2980** |
| | RS | 0.1323 | 0.1719 | 0.1587 |
| | SIIR | 0.1381 | 0.1917 | 0.1738 |
| | SITR | 0.1051 | 0.1325 | 0.1234 |

tests. As shown in Table 12, it can be observed that randomly selecting a fixed set of ICD sequences results in relatively poor performance. The Golden-Set outperforms the best Fix Random set (seed 1) in Avg:1~8 by 16.14. This also demonstrates the importance of high-quality ICD sequences.

## B.3 Fine-grained Analysis of Diverse ICL Methods for VQAv2

Table 13 provides a detailed comparison of different ICL methods on the VQAv2 dataset using IDEFICSv1. We compare the accuracy of different question types in VQAv2 and find Lever-LM consistently outperforms other methods, highlighting its effectiveness in VQA tasks.

Table 14: CIDEr results of different Lever-LM sizes in Image Captioning with IDEFICSv1.

| Model Size | Avg:1~2 | Avg:4~8 | Avg:1~8 |
|---|---|---|---|
| 1-layer Transformer (64.2M) | 89.48 | 107.49 | 101.15 |
| 2-layer Transformer (67.4M) | 89.58 | 105.79 | 100.05 |
| 4-layer Transformer (73.7M) | 89.12 | 106.10 | 100.44 |

Table 15: Accuracy results of diverse ICL methods on SST2 with Qwen1.5-1.8B.

| Method | Avg:1~2 | Avg:4~8 | Avg:1~8 |
|---|---|---|---|
| RS | 0.6227 | 0.6579 | 0.6438 |
| STTR | 0.6807 | 0.7312 | 0.7110 |
| Lever-LM | **0.7087** | **0.7450** | **0.7305** |

Table 16: The inference time of different ICL methods with IDEFICSv1.

| Method | Retrieval Time (s) |
|---|---|
| SIIR | 0.317 |
| Lever-LM | 0.328 |

## B.4 CIDEr Results of Different Lever-LM Sizes in Image Captioning

Table 14 shows the impact of different Lever-LM model sizes on CIDEr scores for image captioning on IDEFICSv1. As shown in Table 14, we evaluate 1-layer/4-layer Transformer decoder layers. We find that the size of Lever-LM has minimal impact on performance. We believe that capturing the ICD sequence distribution is a simple task that can be learned by only a few Transformer Decoder layers. Moreover, our motivation is to use a small model to enhance the ICL performance of a large model, so it is not appropriate to design Lever-LM to be too large.

## B.5 Lever-LM in NLP domain.

To show Lever-LM is a general method, we train a Lever-LM in a NLP task. Specifically, we use Qwen1.5 [65] 1.8B and generate 2-shot ICD datasets for SST-2 [66], which is a sentiment classification task. The accuracy results are displayed in the Table 15. It can be observed that our Lever-LM outperforms the Random method and STTR in Avg:1~2 and Avg: 4~8, demonstrating the potential of Lever-LM in NLP.

## B.6 Inference Time Comparison between Lever-LM and SIIR.

Table 16 compares the inference time between SIIR and Lever-LM methods on IDEFICSv1. Both methods have similar retrieval times, indicating that Lever-LM's performance gains do not come at the cost of efficiency.

## B.7 Performance on VL-ICL Benchmark Tasks

We test Lever-LM on two tasks of VL-bench [67] due to computation limitation and show the results in Table 17. It can be found Lever-LM achieves higher performance than other retrieval-based methods, validating the generalizability of Lever-LM.

## B.8 Performance on IDEFICSv2-8B

We also validate Lever-LM's generalization ability using IDEFICSv2. IDEFICSv2 is an open-source model specifically designed for ICL. Besides, modern LVLMs with robust ICL ability belong to two mainstream architectures : (1) Flamingo-based (using cross-attention to fuse vision and language, like Open-Flamingo or IDEFICSv1 that are tested in our paper) and (2) LLaVA-based (directly concatenating image and language tokens like IDEFICSv2). Thus, testing with IDEFICSv2 assesses Lever-LM's generalization across both architectures. Since IDEFICSv2 directly concatenate vision

Table 17: Results of diverse methods on two tasks from VL-ICL benchmark with IDEFICSv1.

| Task | Method | Avg:1~2 | Avg:4~8 | Avg:1~8 |
|------|--------|---------|---------|---------|
| | RS | 0.145 | 0.209 | 0.188 |
| VL-ICL CLEVR | SIIR | 0.170 | 0.260 | 0.230 |
| | Lever-LM | **0.300** | **0.270** | **0.280** |
| | RS | 0.1923 | 0.230 | 0.218 |
| VL-ICL OCRText | SIIR | 0.155 | 0.164 | 0.161 |
| | Lever-LM | **0.262** | **0.241** | **0.248** |

Table 18: CIDEr score of diverse ICL methods on Image Captioning with IDEFICSv2-8B.

| Method | Avg:1~2 | Avg:3~4 |
|--------|---------|---------|
| SIIR | 81.93 | 99.72 |
| RS | 87.34 | 106.07 |
| Lever-LM | **100.72** | **121.68** |

and text tokens, its input sequence will be longer than IDEFICSv1 and 4-shot inference reaches our GPU limitation, thus we report the results of 1-4 shots. The results in Table 18 show that Lever-LM achieves better results than RS and SIIR. Note that RS outperforms SIIR here, one possible reason is that IDEFICSv2 is more likely be damaged by short-cut inference due to similar ICDs are used, while Lever-LM is more robust.

## C   Fixed Set ICD Configurations.

We present four ICD configurations of the Fixed Set. Figure 4 displays two ICD configurations for IC, and Figure 5 displays the other two ICD configurations for VQA.

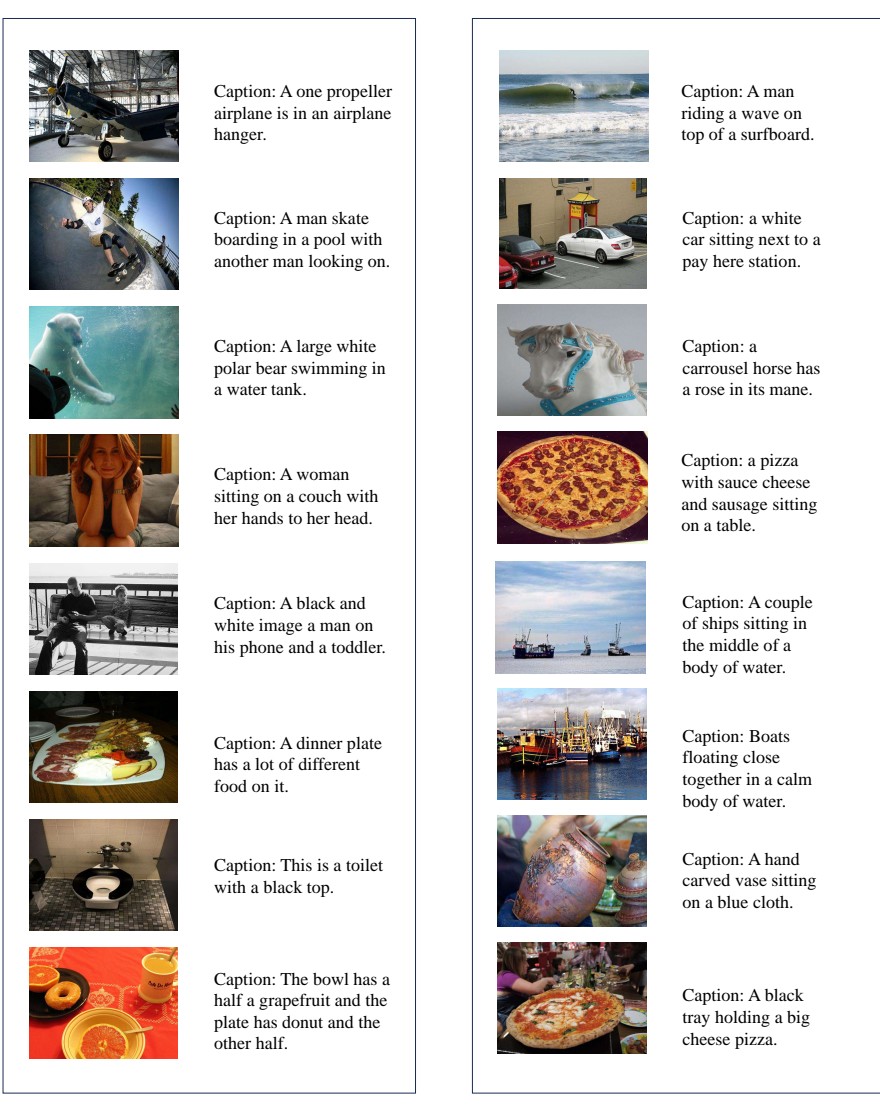

Fixed Set-1: IC                              Fixed Set-2: IC

Figure 4: 8-shot ICD configurations visualizations of IC Fixed Set.

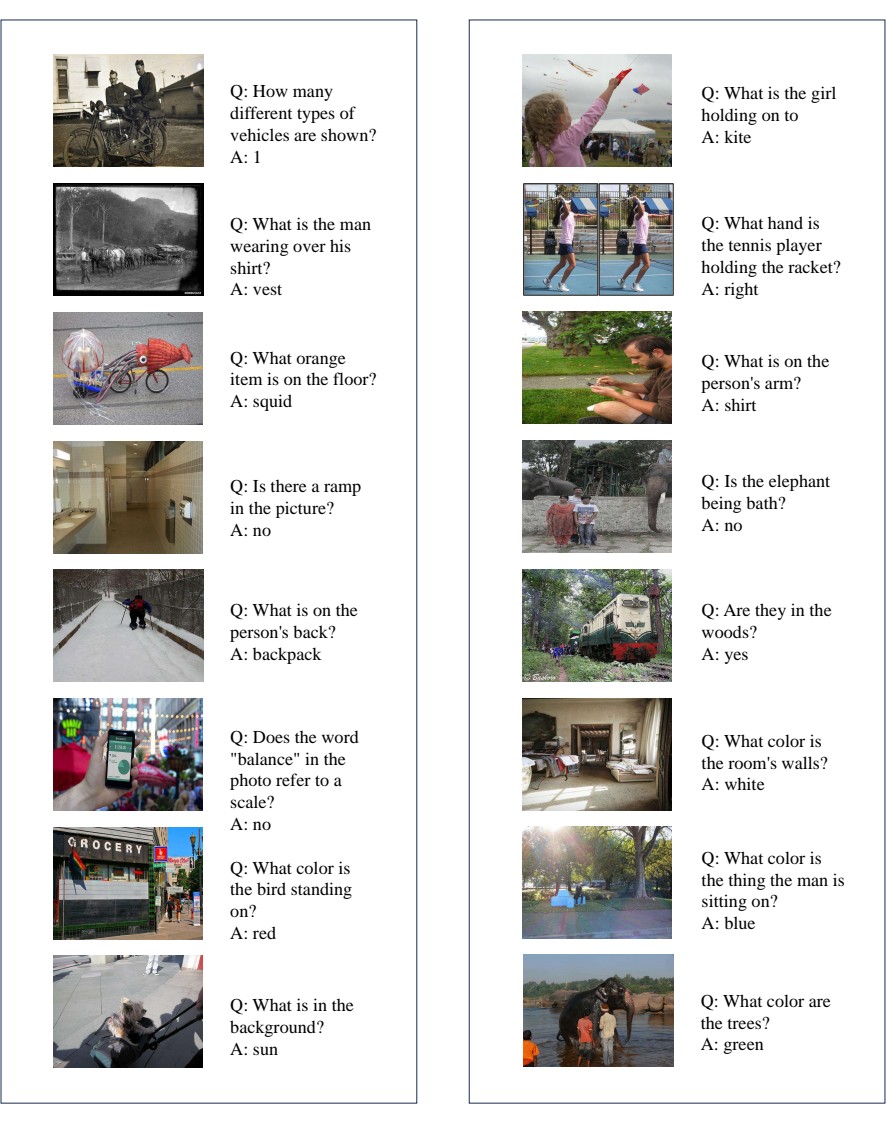

Fixed Set-1: VQA                    Fixed Set-2: VQA

Figure 5: 8-shot ICD configurations visualizations of VQA Fixed Set.

## D  Limitation

One major limitation of our study is the strategy used to build $\mathcal{D}_{\mathcal{M}}$ is not optimal, which requires further improvement. This limitation is revealed by observing that the 4-shot $\mathcal{D}_{\mathcal{M}}$ performs worse than the 2-shot one, highlighting the need for a more effective approach in searching for longer ICD sequences. To address this, we plan to design function $I_{\mathcal{M}}$ to evaluate the effectiveness of ICD sequences; and use better sampling strategies in Eq. (2) to avoid the ICD sequence deviating from the global optimum.

