# OpenReview forum: "Lever LM: Configuring In-Context Sequence to Lever Large Vision Language Models"
_NeurIPS.cc/2024/Conference — NeurIPS 2024 poster_

### Official Review · Reviewer_g8yL · 2024-06-23

**Soundness:** 3
**Presentation:** 3
**Contribution:** 2
**Rating:** 5
**Confidence:** 4

**Summary:**

This paper proposes Lever-LM, a small language model designed to configure effective in-context demonstration (ICD) for improving the in-context learning performance of large vision-language models. The authors construct a dataset of effective ICD sequences to train Lever-LM, which then generates new ICD configurations for novel queries to solve vision-language tasks through in-context learning. Experiments on image captioning and VQA tasks demonstrate that Lever-LM outperforms strong baselines and can capture statistical patterns in ICD sequences to effectively leverage LVLMs.

**Strengths:**

- The paper introduces an interesting method using a small language model (Lever-LM) to configure in-context demonstrations for LVLMs.  Empirical results on image Captioning and VQA demonstrate the effectiveness of Lever-LM across these settings.
- The paper includes a wide range of ablation studies exploring various aspects of the proposed method, such as different configurations for constructing the training dataset, different model architectures, and the impact of ICD sequence length. These studies provide valuable insights into the factors affecting the performance of Lever-LM.
- The paper is easy to understand.

**Weaknesses:**

- In practice, we use ICL because we do not want to update the model parameters or use more computing, while this approach requires training a small model for ICD selection. Thus, the authors need to show that this method can generalize to more models (e.g., Qwen-VL, InternLM-XComposer2, IDEFICS2, GPT4, etc) and new tasks beyond VQA and captioning (e.g., [1]), and therefore show whether the additional computations worth it.
- Zero-shot performance should also be shown as a reference to few-shot performance in Table 1, 2, 3, and more.
- Lever-LM is trained on 2-shot and the authors show the extrapolation to up to 8 shots, and the performance is almost constant wrt. to the number of shots for extrapolation except for OF IC. This may indicate the small number of shots during training makes this strategy hard to generalize to more shots beyond, which limits more performance gain and applications such as many-shot ICL.

[1] VL-ICL Bench: The Devil in the Details of Benchmarking Multimodal In-Context Learning. arXiv, 2024.

**Questions:**

There are many studies arguing about the poor metrics (Rouge, CIDEr, BLEU, etc) of image captioning because the n-gram based metrics give higher scores for captions that have a similar format as the GT. This is especially true for the ICL setting because the model will follow the given ICD, and therefore the predicted caption will just be more similar to the given caption but not necessarily better in terms of accuracy, factuality, etc. Have you tried to use LLM-based evaluation for image captioning?

**Limitations:**

See above.

---

> ### Author Rebuttal · Authors · 2024-08-07
>
> **1.1.Why training auxiliary model.**
>
> Just as you commented, ICL does not need to update the model parameters. However, this refers to LLM or LVLM, not to the auxiliary models used for selecting good ICDs. In fact, NLP researchers have developed various auxiliary methods to retrieve and order ICDs [41,42,44,45], while few studies have done so for VL ICL as of the submission deadline. Our work is a pioneering effort in this area. Furthermore, we are the first to apply the concept of treating ICD generation as a Language Modeling task, a strategy yet unexplored in NLP, to the more challenging VL field.
>
> **1.2. More benchmarks and models.**
> We test Lever-LM on two tasks of VL-bench [A] due to computation limitation and show the results in Table I. It can be found Lever-LM achieves higher performance than other retrieval-based methods, validating the generalizability of Lever-LM. We also follow your suggestion to validate Lever-LM's generalization ability using IDEFICSv2 from the diverse LVLMs you mentioned, due to computational constraints. Our choice is based on two factors. First, IDEFICSv2 is an open-source model specifically designed for ICL. Second, modern LVLMs with robust ICL ability belong to two mainstream architectures : (1) Flamingo-based (using cross-attention to fuse vision and language, like Open-Flamingo or IDEFICSv1 that are tested in our paper) and (2) LLaVA-based (directly concatenating image and language tokens like IDEFICSv2). Thus, testing with IDEFICSv2 assesses Lever-LM's generalization across both architectures. Since IDEFICSv2 directly concatenate vision and text tokens, its input sequence will be longer than IDEFICSv1 and 4-shot inference reaches our GPU limitation, thus we report the results of 1-4 shots. The results in Table J show that Lever-LM achieves better results than RS and SIIR. Note that RS outperforms SIIR here, one possible reason is that IDEFICSv2 is more likely be damaged by short-cut inference due to similar ICDs are used, while Lever-LM is more robust.
>
> To validate our hypothesis, we review examples of text generated by SIIR, such as the following:
> ```
> SIIR Model Output [The first is the most similar ICD, and the last is the query]:
>    Caption:Many children are posing together outside of the building window. [ICD1]
>    Caption:A group of children are sitting together wearing dresses and suits and ties. [ICD2]
>    Caption:Many children are posing together outside of the building window. [Query Prediction]
> Lever-LM Model Ouput:
>    Caption:School children cheer a tennis match with a pirate and giant tennis racket. [ICD1]
>    Caption:A boy in a baseball cap holding baseball mitt.  [ICD2]
>    Caption:A group of school children posing for a photo. [Query Prediction]
>
> Ground Truth
> 1. Many small children are posing together in the black and white photo.
> 2. A vintage school picture of grade school aged children.
> 3. A black and white photo of a group of kids.
> 4. A group of children standing next to each other.
> 5. A group of children standing and sitting beside each other.
> ```
> It is evident that the model directly copied the caption from the first ICD, and such errors are not uncommon. We speculate that the LVLM's coarse visual models cannot distinguish excessively similar images, leading the LVLM to treat the ICD and the query as the same image, resulting in the model copying the ICD's caption.
>
> Additional, we test Lever-LM in NLP ( **A.2.1 of R.RjFC**) and show more fine-grained analyses based on IC and VQA (**A.3.2 of R.sise**). **A.2.2 of R.RjFC** also discusses why Lever-LM is a generalizable ICD configuration strategy in VL.
>
> [A] VL-ICL Bench: The Devil in the Details of Benchmarking Multimodal In-Context Learning
>
> **2.Zero-shot Performance.**
>
> We evaluate the zero-shot performance in Table H, showing that when compared with zero-shot performance, all the ICL methods achieve significantly improvements. We will incorporate zero-shot results in all Tables in revision.
>
> **3.Constant performance of more shots.**
>
> Many-shot ICL performance is influenced by two factors: LVLM's capacity for addressing long input sequences and the in-context sequence quality.
> Studies on Open-Flamingo v2 and IDEFICSv1, Which are two LVLMs used in our experiments, indicate that these LVLMs exhibit limited ability with many-shot ICDs [12][13][29]. Specifically, these LVLMs are trained by 5-shot ICEs, meaning limited capacity for more shot inputs. Our findings, such as Table 1, also show limited performance gains from various retrieval-based methods using more-shot ICDs. Yet, Lever-LM outperforms these methods on average. This does not imply Lever-LM's generalization to more-shot cases is compromised. We appreciate your feedback and plan to test Lever-LM with LVLMs better suited for longer inputs.
>
>
> **4.ClipScore.**
>
> We appreciate your suggestion about calculating the CLIPScore. High-quality captions generated from a model should hinge on two aspects: linguistic correctness and visual congruity. CIDEr and CLIPScore respectively values the former and the latter.
> The CLIPScores of diverse methods are given in Table K. We can find that RS has higher CLIPScore than SIIR, while SIIR has higher CIDEr than RS (Table 1). One possible reason is that when using SIIR, LVLM might copy the captions of ICDs with similar images as the query and overlook the vision content of query image. However, since RS returns less similar in-context images as the query, the short-cut inference is alleviated and thus has higher CLIPScore. Furthermore, we find Lever-LM has the highest CLIPScore, suggesting Lever-LM generates dissimilar ICDs to the query while meantime help LVLM generate the captions which are more grounded to the images. These comparisons validate that Lever-LM can capture useful statistic patterns between high-quality ICDs instead of simply generating similar ICDs.

---

> > ### Comment · Reviewer_g8yL · 2024-08-12
> >
> > Thank you very much for the clarification and additional experiments. I'm raising my score to 5, and hope these discussions/experiments can be integrated into the final version. \
> > I have one additional question: in Table I, why the performance of "Avg:4∼8" is lower than "Avg:1∼2", any intuition?

---

> ### Author Response · Authors · 2024-08-13
>
> We appreciate your response. Due to the time constraints during the rebuttal phase, we only generated 150 high-quality ICD sequence data, which may have led to a decrease in extrapolation ability. After Rebuttal, we continued to generate more data (500 high-quality ICD sequences) and in the VL-ICL benchmark and trained a Lever-LM with it. The results are as follows:
> | Task           | Method   | Avg:1~2 | Avg:4~8 | Avg:1~8 |
> |----------------|----------|---------|---------|---------|
> | VL-ICL clevr   | RS       | 0.145   | 0.209   | 0.188   |
> |                | SIIR     | 0.170   | 0.260   | 0.230   |
> |                | Lever-LM | **0.305**   | **0.31**    | **0.308**   |
> | VL-ICL OCRText | RS       | 0.1923  | 0.230   | 0.218   |
> |                | SIIR     | 0.155   | 0.164   | 0.161   |
> |                | Lever-LM | **0.265**   | **0.274**   | **0.270**   |
>
> The results show that the Lever-LM still has the extrapolation abilities.

---

### Official Review · Reviewer_Djwv · 2024-07-11

**Soundness:** 3
**Presentation:** 3
**Contribution:** 4
**Rating:** 8
**Confidence:** 5

**Summary:**

This paper presents a novel approach called Lever LM, which uses a tiny language model to configure effective ICD sequences for LVLMs. The key innovation is leveraging the small Lever LM to select and order ICDs, improving the ICL performance of LVLMs on tasks like VQA and image captioning. The paper demonstrates that Lever LM captures statistical patterns in ICD configurations, leading to significant performance gains over existing methods.

**Strengths:**

This paper proposes an innovative approach: training Lever LM to select ICD for in-context learning.

Treating ICD selection as a sequence generation task is novel. The experimental results also demonstrate the effectiveness of this modeling approach. Compared to heuristic ICD selection methods, Lever LM demonstrates greater robustness, significantly outperforming other selection methods in ICL across multiple models and tasks, achieving optimal performance.

In addition, the method shows the surprising ability to perform k-shot learning despite being trained in a 2-shot setting. The golden fixed set is an interesting phenomenon, which points that finding golden fixed set may become an important research direction in the future.

**Weaknesses:**

1.	The model needs to be encoded with CLIP first. It's seemed CLIP is also a big model compare than Lever LM.

2.	There is no comparison with traditional ICDs ranking methods.

**Questions:**

1.	Will CLIP model as data encoder affect inference efficiency?

2.	Typo: In Figure 3, what ICD-LM means Lever LM?

---

> ### Author Rebuttal · Authors · 2024-08-07
>
> **1. The role of CLIP**
>
> The role of the CLIP model is to encode the data in the supporting set. Therefore, in practical applications, the dataset can be pre-encoded and stored locally.
>
> **2. Lever-LM Inference Time.**
>
> During inference, only two layers of Transformer decoders are needed, which results in low computational overhead. We show the inference time of diverse methods in Table G. It can be found that Lever-LM does not have a significant gap in inference time compared to the SIIR method.
>
>
> **2. Compared with other Ranking Methods.**
>
> In Line 102, we mention that the ordering of ICD sequences has been primarily studied within NLP [17,24]. Their methods are mostly tested on datasets with limited labels (e.g., binary sentiment classification) or require calculating the probability of each input token. Transferring these methods to VL tasks presents two challenges: 1. VL tasks often include continuous image features, and modeling the probability distribution of continuous features is difficult. 2. VL tasks like IC or VQA are often open-ended word generation tasks, which do not have a limited label space. Facing these challenges, there is limited study explores how to order VL ICDs for improving ICL performance. In fact, we are the first to model the ordering of VL ICD sequences up until the submission deadline, thus it is hard to find suitable comparable methods. Table 4 demonstrates that when the quality of the selected ICDs is the same, the order produced by Lever-LM is optimal, indicating that Lever-LM can generate the proper order.
>
> **3. Typo in Figure 3.**
> We apologize for the typo, and we will revise this part in revision.

---

> > ### Comment · Reviewer_Djwv · 2024-08-11
> > **Official Comment by Reviewer Djwv**
> >
> > Thanks for the response. I also roughly browse the comments of the other reviewers and the corresponding feed. After reading the response to Reviewer g8yL, I agree with his comment that Image Captioning is an ill-posed task and the main reason is the n-gram based metric, while I think VQA is still a suitable task for evaluating the effectiveness. Also, it is glad to see the additional experiments on other benchmarks and the results on CLIPScore.
> > I have a few additional questions related to the results on IDEFICs v2. From the results in Table J, the authors still report the performance by CIDEr, what about the CLIPScore? Also, it is interesting to find that IDEFICs v2 may directly copy the captions in the ICDs, can the authors provide more discussions about this?

---

> > > ### Author Response · Authors · 2024-08-13
> > >
> > > Thanks for this question. We follow your suggestion to test CLIPScore and CHs/CHi scores to measure hallucinations. The results averaged on 1 to 8-shot are given as follows:
> > >
> > > |           | Lever-LM | RS    | SIIR  |
> > > |-----------|----------|-------|-------|
> > > | CLIPScore | **0.782**    | 0.770 | 0.761 |
> > > | CHs       | **5.48**     | 6.08  | 7.5   |
> > > | CHi       | **4.56**     | 5.04  | 5.7   |
> > >
> > > For IDEFICSv2, since its vision encoder can only grasp global semantics, and a high similarity in images often may imply that the features encoded by the LVLM are also similar, leading to a tendency for the LVLM to produce identical outputs when applying similar-based retrieval methods like SIIR. However, Lever-LM is more robust that it does not generate the similar images as the query one, and thus alleviating the hallucinations. Here we show a few more examples to demonstrate this.
> > >
> > > ### Example 1
> > >
> > > SIIR: **A bowl filled with fruit and vegetables on a counter.** [ICD1] A pile of bananas, oranges and pears sitting next to each other. [ICD2] **A bowl filled with fruit and vegetables on a counter.** [Query]
> > >
> > > LeverLM: A red two story bus drives down the street. [ICD1] A woman wearing a short black skirt while holding a tennis racquet. [ICD2] A black and white drawing of fruits and vegetables. [Query]
> > >
> > > Ground Truth Captions:
> > > - A black and white image of a lot of round objects.
> > > - Two apples, an orange, some grapes and peanuts.
> > > - A black and white photo of nuts and fruit.
> > > - A pile of nuts in front of some assorted fruit.
> > > - Apples, grapes, an oranges and peanuts on the white surface in a picture.
> > >
> > >
> > > ---
> > >
> > > ### Example 2
> > >
> > > SIIR: A little boy stands outdoors on a rainy day with a pink umbrella. [ICD1] **A little girl that is in the grass with an umbrella.** [ICD2] **A little girl that is in the grass with an umbrella.** [Query]
> > >
> > > LeverLM: Two airplanes sitting on top of an airport tarmac. [ICD1] A man is smiling while sitting on a horse. [ICD2] A young girl is holding an umbrella. [Query]
> > >
> > > Ground Truth Captions:
> > > - A little girl holds up a big blue umbrella.
> > > - A young girl stands with her arms wrapped around a large blue umbrella.
> > > - A girl in a pink shirt holding a blue umbrella.
> > > - A little girl who is holding an umbrella.
> > > - A little girl with a big, blue umbrella.
> > >
> > >
> > > ---
> > >
> > > ### Example 3
> > >
> > > SIIR: A traffic light sitting on the side of a road. [ICD1] **A traffic light sitting above a white sign.** [ICD2] **A traffic light sitting above a white sign.** [Query]
> > >
> > > LeverLM: A woman holding two different items of food in her hands. [ICD1] A boy in a baseball cap holding baseball mitt. [ICD2] A man standing in front of a traffic light. [Query]
> > >
> > > Ground Truth Captions:
> > > - A man standing next to a light and a sign.
> > > - A man standing next to a traffic light in Australia.
> > > - A man that is standing next to a traffic light.
> > > - A man standing in front of a sign under a street light.
> > > - A man in shorts is taking a picture next to a red light.

---

> > > > ### Comment · Reviewer_Djwv · 2024-08-13
> > > > **Official Comment by Reviewer Djwv**
> > > >
> > > > Ok, I am willing to consider upgrading the rating based on the responses provided by the authors.

---

### Official Review · Reviewer_RjFC · 2024-07-11

**Soundness:** 3
**Presentation:** 3
**Contribution:** 3
**Rating:** 8
**Confidence:** 4

**Summary:**

This paper proposes using a Tiny Lever-LM to assist in ICD selection for LVLM's ICL scenarios, thereby enhancing ICL performance without significantly increasing computational costs. Lever-LM unifies the modeling of multiple scenarios (VQA, IC) in complex multimodal ICL, eliminating the need for manually designed heuristic ICD selection strategies. Additionally, Lever-LM jointly learns ICD selection and ordering, achieving end-to-end learning of ICD sequences. Lever-LM achieves excellent performance across multiple LVLM models, significantly outperforming existing multimodal ICD heuristic selection methods.

**Strengths:**

1. Lever-LM's structure uses only two layers of Transformer Decoder, making it a highly efficient model for generating ICD sequences.

2. Lever-LM unifies the modeling of complex multimodal ICL scenarios, eliminating the need for manually designed heuristic ICD selection strategies. Furthermore, Lever-LM simultaneously models both ICD selection and ordering steps, achieving end-to-end optimization of ICD sequence modeling. Besides, It is the first attempt to establish the ICD selection and sorting task as a generation task.

3. Lever-LM demonstrates robustness on multiple levels. First, the authors test it across various tasks and models, consistently outperforming the best ICD selection methods in the current multimodal context. Meanwhile, manually designed heuristic ICD selection strategies show significant performance fluctuations across different models. Additionally, they evaluate the use of different metrics for constructing ICD sequence datasets and experiment with different language model architectures, all of which show excellent performance.

4. The authors validate that the sequence order generated by Lever-LM performs better than randomly generated sequence orders, proving that Lever-LM indeed learns the order information of ICD sequences.

5. Lever-LM demonstrates interesting length extrapolation ability after being trained on a 2-shot ICD sequence dataset. It performs strongly even when generating 4-shot, 6-shot, and 8-shot sequences. This proves the efficiency of the Lever LM method: it requires only short-shot ICD sequences to exhibit excellent performance on long-shot ICL.

**Weaknesses:**

1. It has not been explored whether the performance of Lever-LM strongly depends on its size.

2. It seems like LLMs can also use this method for ICL tasks. It may be better to evaluate Lever-LM in LLMs to demonstrate its versatility.

**Questions:**

1. How would a smaller or larger Lever-LM affect the results?

2. Can you show the input format of LVLM when doing ICL? Do Idefics will use the same format of OF?

3. Can you evaluate Lever-LM in LLM? I believe it can better demonstrate the generalizability of the method.

4. Compared with RS/SIIR methods, how about the inference time of Lever-LM

**Limitations:**

Please see the Weaknesses.

---

> ### Author Rebuttal · Authors · 2024-08-07
>
> **1. Lever-LM with Different Sizes.**
>
> We follow your suggestion to obtain different sizes of Lever-LM by controlling the number of Transformer layers and test these Lever-LMs on the IC. As shown in Table E, we evaluate 1-layer/4-layer Transformer decoder layers. We find that the size of Lever-LM has minimal impact on performance. We believe that capturing the ICD sequence distribution is a simple task that can be learned by only a few Transformer Decoder layers. Moreover, our motivation is to use a small model to enhance the ICL performance of a large model, so it is not appropriate to design Lever-LM to be too large.
>
>
> **2.1 Lever-LM in NLP.**
>
> We follow your suggestion to train a Lever-LM in NLP. Specifically, we use Qwen1.5 [A] 1.8B and generate 2-shot ICD datasets for SST-2 [B], which is a sentiment classification task. The accuracy results are displayed in the Table F. It can be observed that our Lever-LM outperforms the Random method and STTR in Avg:1\~2 and Avg: 4\~8, demonstrating the potential of Lever-LM in NLP.
>
> [A] Qwen Technical Report
>
> [B] Recursive Deep Models for Semantic Compositionality Over a Sentiment Treebank
>
> **2.2. Generality in VL.**
>
> Our approach is actually a generalizable method in VL. Different VL tasks require diverse configuration strategies. For example, in IC, [20] proposes that the image and caption need to be matched. In VQA, [21] proposes the best ICD selection strategy involves sorting based on image similarity first and then question similarity. In other words, LVLM requires different heuristic ICD selection strategies for different tasks, and it is difficult to find a universal and effective configuration strategy. Our method, on the other hand, provides a generalizable solution that can directly adapt to different models and tasks.
>
> **3. The format of IC and VQA.**
>
> We have detailed the prompt template in A.3.Prompt Template of Appendix. Our prompt template is entirely based on the settings described in the Open-Flamingo and IDEFICS.
>
> **4. The Inference Time of Lever-LM.**
>
> Please refer to **A.2 of R.Djwv** where we show the inference time of Lever-LM and SIIR.

---

> > ### Comment · Reviewer_RjFC · 2024-08-10
> >
> > I appreciate your response and the inclusion of new experiments regarding Lever-LM sizes, new benchmarks, and NLP tasks, which have addressed my concerns. However, I have one further question: Lever-LM appears to learn more abstract knowledge compared to a conventional machine learning model such as an image classifier. How should we interpret this model in relation to more standard ones?

---

> > > ### Author Response · Authors · 2024-08-11
> > >
> > > This is a good question and the following shows my thinking. Traditional machine learning models, which learn specific patterns from human-annotated datasets, can be seen as first-order learning problems. Conversely, Lever-LM represents a second-order learning problem, as it acquires potentially abstract knowledge from a model previously trained on a human-annotated dataset. Prior to the proposal of LLMs, research emphasis was placed on training models to resolve specific tasks like image captioning or visual question answering, with the task serving as the primary research subject. However, the proposal of LLMs has shifted focus towards understanding their inherent characteristics, thereby making these models the new subjects of study. This shift led to the discovery of numerous emergent properties in LLMs, such as prompt engineering and in-context learning. Using these emergent properties as a foundation, researchers employed statistical observation methods to investigate internal characteristics of LLMs, including attention flow [A] and patterns between different layer representations [B]. A parallel research trajectory can be observed in the field of in-context learning. Initial heuristic methods examined the external influence of different ICDs on ICL [C], which subsequently transitioned to statistical observations of the internal characteristics when LLMs perform ICL [D]. Lever-LM aligns with this approach, using a model to discern the internal statistical characteristics of a large model, which suggests that though these characteristics may remain unknown to humans, they objectively exist.
> > > [A] Quantifying attention flow in transformers
> > > [B] ShortGPT: Layers in Large Language Models are More Redundant Than You Expect
> > > [C] Exploring diverse in-context configurations for image captioning
> > > [D] Label Words are Anchors: An Information Flow Perspective for Understanding In-Context Learning

---

> > > > ### Comment · Reviewer_RjFC · 2024-08-12
> > > >
> > > > Thanks for sharing your thinking about LLMs, which inspires me a lot. I think the this shift in research focus is similar to the transition from using heuristic statistical methods to extract image features to adopting learned approaches for learning representations.

---

### Official Review · Reviewer_sise · 2024-07-12

**Soundness:** 2
**Presentation:** 2
**Contribution:** 2
**Rating:** 3
**Confidence:** 5

**Summary:**

The authors focus on configuring effective in-context demonstration (ICD) sequences to improve the In-Context Learinng (ICL) performance of LVLMs. The proposed Lever-LM enables the step-by-step generation of ICD configurations and simultaneously considers the selection of ICDs and the ordering of ICD sequences. Experimental results validate that Lever-LM can capture the statistical patterns for levering LVLMs compared with similarity-based retrieval methods.

**Strengths:**

The tokens of the vocabulary in Lever-LM are the samples from the supporting set. Given the query sample, the ICDs can be selected one by one based on the token distribution produced by the trained Lever-LM.

**Weaknesses:**

1. In lines 196-197, what is the reason for “randomly” choosing samples to build the sub-supporting set? I think this method is suboptimal, personally.
2. Table 2 (15) uses fixed ICD configuration for any query input, and still returns significantly improved performance compared with other methods in IC. I think this phenomenon shows that the ICL method fails to work on IC. Or I am curious about the performance of “randomly” selected fixed ICD configuration compared with model selected.
3. The experiments part is not convincing in the field of MLLMs. I hope to see results with more benchmarks and models (e.g., M-ICL[1]). To assess the generalization capability of their approach, it would be advantageous for the authors to evaluate their methodology using benchmarks that emphasize fine-grained analysis, such as MM-Vet and SEED-Bench, particularly focusing on segments that require in-depth evaluation.
4. I think the performance between two order strategies are comparative in Table 4, personally.

[1] What Makes Multimodal In-Context Learning Work?, CVPR 2024.

**Questions:**

More comparisons with recent MLLMs are required.

**Limitations:**

See weakness

---

> ### Author Rebuttal · Authors · 2024-08-07
>
> **1. Random Sampling.**
>
> The goal of randomly selecting samples to form the sub-supporting set $\mathcal{D_{\mathcal{M}}}$ is to enhance training data diversity, promoting Lever-LM to capture complementary knowledge among ICDs. We initially deemed this to be a strategy sub-optimal, exploring alternatives like selecting similar texts and images (Lines 265-269). Surprisingly, Table 2 (9)-(11) shows random sampling as most effective. This finding (Lines 288-291) suggests that using similar ICDs to form $\mathcal{D_{\mathcal{M}}}$ may make Lever-LM generated ICDs contain redundant knowledge, hindering ICL performance. This also aligns with multiple studies underscoring the benefits of diversity for ICL performance [43,54,A,B].
>
> [A] Diversity of Thought Improves Reasoning Abilities of LLMs
>
> [B] In-Context Learning with Iterative Demonstration Selection
>
> **2.Fixed ICD Configuration.**
>
> Actually, the existence of a good fixed set for IC proves that ICL works on IC. This is because ICL's main goal is to facilitate efficient learning of new tasks with minimal examples. To achieve this, one ongoing research direction in ICL is to find a minimal supporting set for ICD selection [C,D], and our discovery of a "golden fixed set" requiring only 8 ICDs is an extreme instance of this direction. Hence, our findings robustly suggest ICL's potential for IC.
>
> We also follow your suggestion to compare the fixed set learned by Lever-LM with the one constructed by random selection.
> Specifically, we use OpenFlamingo to randomly select 3 sets of k-shot ICD sequences with different seeds for the image captioning task and then conduct ICL tests. As shown in Table A in the rebuttal PDF, it can be observed that randomly selecting a fixed set of ICD sequences results in relatively poor performance. The Golden-Set outperforms the best Fix Random set (seed 1) in Avg:1~8 by 16.14 points. This also demonstrates the importance of high-quality ICD sequences.
>
> [C] Finding Support Examples for In-Context Learning
>
> [D] Compositional Exemplars for In-context Learning
>
>
> **3.1.Why VQA and IC.**
>
> It should be stressed that our experimental settings are different from M-ICL [E] that only needs to forward LVLM for ICL inference, while we need to train Lever-LM by modifying various settings (Lines 262-292) to identify which factors influence the learning. Given limited resources, we focus on essential vision-language tasks, IC and VQA, to evaluate Lever-LM. IC and VQA also play important roles in LVLM ICL. First, they are commonly used for evaluating LVLM's ICL performance [12,13], even are specifically studied to help understand the LVLM ICL abilities [20,21]. Also, IC and VQA incorporate diverse vision-language skills: IC tests object, attribute, and relation recognition, while VQA covers diverse question types, integrating various capabilities like classification ('what is the object?') or counting ('how many of one object?'). Modern benchmarks for evaluating LVLM often derive from the IC (COCO dataset) and VQA (VQAv2 dataset). For example, SEED-Bench [F], MME [G] and MMBench [H] incorporate core characteristics from IC and VQA tasks, assessing various aspects such as scene understanding, instance identity, spatial relations, and diverse question types.
>
> [E] What Makes Multimodal In-Context Learning Work?
>
> [F] SEED-Bench: Benchmarking Multimodal LLMs with Generative Comprehension
>
> [G] MME: A Comprehensive Evaluation Benchmark for Multimodal Large Language Models
>
> [H] MMBench: Is Your Multi-modal Model an All-around Player?
>
> **3.2.Fine-grained Analyses.**
>
> We value your fine-grained analysis suggestions but find MM-Vet [I] and SEED-Bench not fitting our study. MM-Vet's limited scale (200 images, 218 questions) is not sufficient for effective Lever-LM training. SEED-Bench, designed for multiple-choice questions, does not match well with LVLMs that focus on word generation in ICL, requiring specific adaptations that are challenging to us given the rebuttal's time constraints. However, as previously mentioned, VQA and IC are appropriate for fine-grained analysis. For IC, we assess object hallucination using CHAIR [J], providing object-level results, where larger CH$_s$ and CH$_i$ values indicate more object hallucinations. As Table B\&C shows, Lever-LM ICDs, compared to random sampling-based and similarity-based retrieval methods, achieve smaller hallucination scores because they contain fewer similar images to the query, preventing LVLM from short-cut learning. For VQA, we calculate the accuracy of three question types for fine-grained analyses: `Yes/No`, `Counting`, `Others` (Table D), which shows that Lever-LM significantly outperforms other methods in `Counting`, suggesting Lever-LM ICDs helps LVLM get stronger fine-grained object recognition ability.
>
> [I] MM-Vet: Evaluating Large Multimodal Models for Integrated Capabilities
>
> [J] Object Hallucination in Image Captioning
>
> **3.3.More Models and Benchmarks.**
>
> We follow your suggestion to test Lever-LM on more models and benchmarks. Specifically, we examine the generalizability in VL-ICL-Bench (**A.1.2 of R.BDyQ**),  Qwen1.5 model in NLP (**A.2.1 of R.RjFC**), and by another LVLM architecture, IDEFICSv2 (**A.1.2 of R.BDyQ**).
>
>
> **4.Order strategies.**
>
> The comparative results in Table 4 primarily stem from Lever-LM unifying the generation and ordering of ICDs. Lever-LM generates ICDs with complementary knowledge, enhancing the overall performance and diminishing the importance of ICD ordering. Thus, it is hard to mirror the significant improvements as the studies focusing solely on ICD ordering. Yet, Table 4 shows that Lever-LM's generated order outperforms random order given high-quality generated ICDs, demonstrating its effectiveness in ordering ICDs.

---

### Author Rebuttal · Authors · 2024-08-07

We gratefully thank all the reviewers for their valuable and constructive feedback. We are pleased to see that the reviewers recognize our motivation: to use a tiny model to enhance the performance of in-context learning (ICL) for LVLMs. We are encouraged to see that they find our method is novel, interesting (Reviewer Djwv, g8yL, RjFC), our method is a unified method for ICL in NLP and Visual Language domain (Reviewer RjFC), our experiment is very detailed and effective (Reviewer g8yL, Djwv, sise, RjFC).


We address the concerns and questions in detail below and have appended a PDF file with tables. **Note that to distinguish from the Tables and References in the submitted manuscript, the Table and Reference numbers in the rebuttal are marked with A, B, C, etc.**
Based on these comments, we have summarized the common questions and our responses as follows:

1. We conduct supplementary experiments on other tasks and LVLMs to demonstrate the general applicability of Lever-LM (Table F/I/J, to Reviewers sise, RjFC, g8yL).


2. We provide more analysis of Lever-LM, including a fine-grain analysis of VQA and IC and another metric to evaluate different ICL methods' performance on IC (Table B/C/D/K, Reviewer sise, BDyQ).


3. We show the inference time of Lever-LM and one retrieval-based method to show that the Lever-LM will not influence the LVLMs inference time (Table G, Reviewer RjFC, Djwv).

We also address other specific concerns in separate responses.

---

### Decision · Program_Chairs · 2024-09-25

**Decision:**

Accept (poster)

**Comment:**

This work proposes training a small language model (couple of transformer layers) with "golden" in-context demonstration (ICDs) sequences, in order to (1) select, and (2) re-order in-context demonstration sequences for new queries. The ICDs so-produced are then input into vision language models (VLMs) for in-context learning.

This work generally received positive reviews, with reviewers appreciating:
(1) using the small-LM as an innovative/interesting approach for improving in-context learning.
(2) detailed experimentations, and ablations, demonstrating the proposed method to be robust to: (1) various tasks, (2) models, (3) number of examples, and (4) also applicable to LLMs / NLP tasks.

There are concerns regarding: extra-compute, which in-context learning in principle seeks to avoid, e.g., adaptation without fine-tuning, and requiring a sizeable dataset of in-context +/ golden examples. Further, no comparisons to other (learnt) ICD selection/re-ordering methods is provided.

I urge the authors to include the experiments presented in the review discussions in their paper. Further, experiments addressing the weaknesses highlighted above would help strengthen the paper: e.g., comparison against directly fine-tuning the VLM with the same compute budget and data, and, comparisons with other ICD selection/re-ordering methods from NLP -- as the method has been demonstrated to be applicable to LLMs as well (in Table F).